

# The importance of molecular characters when morphological variability hinders diagnosability: systematics of the moon jellyfish genus *Aurelia* (Cnidaria: Scyphozoa)

Jonathan W. Lawley[1,2], Edgar Gamero-Mora[1], Maximiliano M. Maronna[1], Luciano M. Chiaverano[3], Sérgio N. Stampar[4], Russell R. Hopcroft[5], Allen G. Collins[6] and André C. Morandini[1,7]

[1] Departamento de Zoologia, Instituto de Biociências, Universidade de São Paulo, São Paulo, São Paulo, Brazil
[2] School of Environment and Science, Coastal and Marine Research Centre, Australian Rivers Institute, Griffith University, Gold Coast, Queensland, Australia
[3] Instituto Nacional de Investigación y Desarrollo Pesquero, Mar del Plata, Buenos Aires, Argentina
[4] Departamento de Ciências Biológicas, Faculdade de Ciências e Letras, Universidade Estadual Paulista, Assis, São Paulo, Brazil
[5] College of Fisheries and Ocean Sciences, University of Alaska—Fairbanks, Fairbanks, Alaska, United States
[6] National Systematics Laboratory of the National Oceanic and Atmospheric Administration Fisheries Service, National Museum of Natural History, Smithsonian Institution, Washington, District of Columbia, United States
[7] Centro de Biologia Marinha, Universidade de São Paulo, São Sebastião, São Paulo, Brazil

Corresponding author
Jonathan W. Lawley,
jonathan.wanderleylawley@griffithuni.edu.au

## ABSTRACT

Cryptic species have been detected across Metazoa, and while no apparent morphological features distinguish them, it should not impede taxonomists from formal descriptions. We accepted this challenge for the jellyfish genus *Aurelia*, which has a long and confusing taxonomic history. We demonstrate that morphological variability in *Aurelia* medusae overlaps across very distant geographic localities. Even though some morphological features seem responsible for most of the variation, regional geographic patterns of dissimilarities are lacking. This is further emphasized by morphological differences found when comparing lab-cultured *Aurelia coerulea* medusae with the diagnostic features in its recent redescription. Previous studies have also highlighted the difficulties in distinguishing *Aurelia* polyps and ephyrae, and their morphological plasticity. Therefore, mostly based on genetic data, we recognize 28 species of *Aurelia*, of which seven were already described, 10 are formally described herein, four are resurrected and seven remain undescribed. We present diagnostic genetic characters for all species and designate type materials for newly described and some resurrected species. Recognizing moon jellyfish diversity with formal names is vital for conservation efforts and other studies. This work clarifies the practical implications of molecular genetic data as diagnostic characters, and sheds light on the patterns and processes that generate crypsis.

# INTRODUCTION

## Challenges in 21st century taxonomy

As genetic datasets necessary for species delimitation have grown, "integrative taxonomy" that explicitly incorporates molecular data (*Dayrat, 2005*) is becoming more common. Yet, taxonomy has always been integrative, relying on a broad range of available data that expand with the advent of new technologies, and it is no surprise that derived biological information should also be included (*Valdecasas, Williams & Wheeler, 2008*). That said, it is important to note that integration should be conducted with thoughtfulness and rigor in order to assess the utility of different lines of evidence and their use for constructing species hypotheses (*Valdecasas, Williams & Wheeler, 2008*; *Conix, 2018*). For example, DNA barcoding has been proposed as a tool for rapid species identification (*Hebert et al., 2003*, *2004*), but the term 'species identification' has not had a standard meaning in the barcoding literature (*Goldstein & DeSalle, 2011*; *Collins & Cruickshank, 2013*; *DeSalle & Goldstein, 2019*). Further issues relate to adherence of arbitrary distance thresholds (*Wiemers & Fiedler, 2007*), incomplete reference databases, misidentified sequences and more (*Collins & Cruickshank, 2013*; *Ransome et al., 2017*; *Paz & Rinkevich, 2020*). In spite of these complications, improvements have been made to optimize identification thresholds and assess their viability for certain groups (*Virgilio et al., 2012*), and it cannot be denied that molecular data have proven vital to the discovery of cryptic species, which are two or more distinct species previously unrecognized due to apparent or real morphological resemblance (*Bickford et al., 2007*).

Cryptic species seem to occur across all metazoan groups and biogeographic zones, although some studies have suggested phylogenetic and ecological patterns in the distribution of this phenomenon (*Bickford et al., 2007*; *Pfenninger & Schwenk, 2007*). However, it is questionable whether cryptic species have been studied thoroughly and broadly enough across taxa to confidently assert such patterns (*Trontelj & Fišer, 2009*). Indeed, it seems overly simplistic to generalize about cryptic species diversity at the level of phylum, as there is an astounding variety of speciation-related processes that occur at the genus level (*Trontelj & Fišer, 2009*; see *Coyne & Orr, 2004*). Lack of morphological characters that can adequately distinguish species should be researched further to deepen understanding of morphological variation and the scope and scale of cryptic diversity. The challenge is particularly acute when one shifts from species delimitation to species description. Recognizing real rather than apparent diversity is essential not only for conservation efforts to define priorities and avoid local extinctions, but also for understanding patterns of, and processes that generate, crypsis (see review in *Bickford et al., 2007* and further discussions in *Struck et al., 2017*).

## From species delimitation to description

Taxonomy remains incomplete if discovered entities are not formally described, and species hypotheses are flagged as merely putative, creating parallel worlds populated by numbered candidate taxa (*Jörger & Schrödl, 2013*). The collapse of taxonomic expertise creates a sense of urgency and increasing reliance on molecular data as the solution for sustainable identification (*Hebert et al., 2003*). If that is the case, we must reconcile the precise mechanics of these data with the empirical and philosophical rigor of systematics and taxonomy (*Goldstein & DeSalle, 2011*). Without formal descriptions and testable hypotheses represented by unambiguous names, discovered species are not properly documented, and often not associated to vouchered specimens deposited at museums. Confusion arises from informal naming or numbering schemes of detected lineages when formal taxonomic practice is side-stepped, thereby limiting biodiversity analyses (*Jörger & Schrödl, 2013*; *Pleijel et al., 2008*). Many attempts have been made to incorporate DNA sequence information in taxonomic descriptions, such as including GenBank accession numbers, DNA barcode sequences, raw distance measures and phenetic or phylogenetic trees, but rarely are diagnostic sequence characters identified (see reviews and operational terminology in *Goldstein & DeSalle, 2011*, *DeSalle & Goldstein, 2019*). Nevertheless, a consensus view holds that species descriptions should be character-based (*Bauer et al., 2011*).

Even though it may be artificial to assume that the biological reality of a species depends on a number of diagnostic characters, it provides a falsifiable and comparable framework in which to construct and assess species hypotheses (*Grant et al., 2006*; *Bauer et al., 2011*). Furthermore, the inclusion of diagnostic characters is required for new species names by the International Code of Zoological Nomenclature (*ICZN (International Commission on Zoological Nomenclature), 1999*, Article 13.1.1., also see definition for 'character' in its Glossary). There are now computational tools that can provide diagnostic molecular genetic characters, such as CAOS (*Sarkar, Planet & DeSalle, 2008*), YBYRÁ (*Machado, 2015*), QUIDDICH (*Kühn & Haase, 2019*) and DeSignate (*Hütter et al., 2020*). Even though these programs compile and evaluate diagnostic characters under different strategies, which are yet to be rigorously assessed, they provide a basis for the description of cryptic species. Some efforts have already been made to describe cryptic species (*Jörger & Schrödl, 2013*; *Delić et al., 2017*), and even though there is an ongoing increase in recognition of the economic and ecological importance of scyphozoan jellyfishes (*Hamilton, 2016*; *Hays, Doyle & Houghton, 2018*), a relative lack of investment still results in a taxonomic impediment today (*Gómez-Daglio & Dawson, 2019*).

## A scyphozoan jellyfish with a long taxonomic history

In the moon jellyfish genus *Aurelia*, the subject of the present study, taxonomic history dates back to the 18th century, starting with the description of the type species *Aurelia aurita* (*Linnaeus, 1758*). Since then, this genus has encompassed as many as eight (*Haeckel, 1880*), 13 (*Mayer, 1910*, considering varieties) or seven accepted species (*Kramp, 1961*). More recently, only two species were recognized, *Aurelia limbata Brandt, 1835*, which has a brown bell margin and is primarily from temperate regions, and a cosmopolitan

nearshore inhabitant *A. aurita*, which included as synonyms most of the previously proposed names (*Larson, 1990*; *Arai, 1997*). In the 2000s, two species (*Aurelia labiata Chamisso & Eysenhardt, 1821* and *Aurelia marginalis Agassiz, 1862*) were resurrected based on morphological and geographical differences (*Gershwin, 2001*; *Calder, 2009*), and with the inclusion of genetic data, there were indications of at least another 16 species, some of which are hypothesized to have been introduced in several localities (*Dawson & Jacobs, 2001*; *Schroth et al., 2002*; *Dawson, Gupta & England, 2005*; *Gómez-Daglio & Dawson, 2017*). The most recent accounts of valid described species is up to 10 (*Jarms & Morandini, 2019*; *Collins, Jarms & Morandini, 2020*).

In addition to the evaluation of genetic data, other studies have also reassessed morphological features, taking into consideration morphometric data and not only in the medusa stage (*Dawson, 2003*), but also in other life cycle stages, such as polyps and ephyrae (*Gambill & Jarms, 2014*). A recent result of the integration of these morphological reassessments with genetic data delimited and described or redescribed three species that occur in the Mediterranean, *Aurelia coerulea von Lendenfeld, 1884*, *Aurelia relicta* (*Scorrano et al., 2016*) and *Aurelia solida Browne, 1905* (*Scorrano et al., 2016*). However, some of the reported diagnostic morphological features seemed to overlap across these species, and morphological variability was shown to be considerable, especially in the redescribed *A. coerulea* (see Fig. 6 in *Scorrano et al., 2016*). Other findings further highlight the widespread potential for high morphological variability, such as ecophenotypic plasticity in *Aurelia* medusae (*Chiaverano, Bayha & Graham, 2016*), as well as in the other stages of the life cycle (*Chiaverano & Graham, 2017*).

In the present study, we re-examine the use of morphological data in *Aurelia* medusae, the most conspicuous and collected of the life cycle stages, as well as present a molecular genetic phylogeny for the genus, based on mitochondrial and nuclear markers. In addition, we evaluate previous morphological diagnoses proposed for *Aurelia* species. In combination with recorded geographic distributions, this dataset provides a framework to delimit and describe species, as well as identify new geographical occurrences and potential introductions. With this study, we hope to encourage the transition from species delimitation to description, advance discussions on theoretical and practical applications related to diagnosis from molecular data as part of taxonomy, and expand perspectives for morphological studies to address questions regarding morphological variability, ecophenotypic plasticity, and the evolution of cryptic diversity.

## MATERIALS & METHODS

### Morphological data collection

Observations were made on living medusae from aquariums in the USA, and on preserved medusae from museums and universities from the USA, Brazil and Denmark. In total, 173 specimens were analyzed (Table 1; specimen vouchers, identification, sampling sites and correspondence to molecular data of specimens observed are provided in Tables S1–S2). Two live medusae from laboratory cultures at the Laboratory for Cnidarian Studies and Cultivation (University of São Paulo), identified from genetic analyses as *Aurelia coerulea* (presented further), were also used, but only for the purpose of a

**Table 1 Institutions from which specimens were observed for morphological analyses and included in this study.**

| Institution | City, Province | Country | Live/Preserved | N |
|---|---|---|---|---|
| Smithsonian Institution's National Museum of Natural History (USNM) | Washington, DC | USA | Preserved | 81 |
| Zoological Museum of the University of Copenhagen (ZMUC) | Copenhagen | Denmark | Preserved | 25 |
| Yale Peabody Museum of Natural History (YPM) | New Haven, CT | USA | Preserved | 24 |
| Laboratory for Cnidarian Studies and Cultivation of the University of São Paulo (LAB) | São Paulo, SP | Brazil | Preserved | 20 |
| Discovery Place (DP) | Charlotte, NC | USA | Live | 8 |
| American Museum of Natural History (AMNH) | New York, NY | USA | Preserved | 5 |
| Federal University of Bahia (UFBA/MZUFBA) | Salvador, BA | Brazil | Preserved | 3 |
| Museum of Zoology of the University of São Paulo (MZUSP) | São Paulo, SP | Brazil | Preserved | 3 |
| National Aquarium (NA) | Baltimore, MD | USA | Live | 2 |
| Federal University of Ceará (UFC) | Fortaleza, CE | Brazil | Preserved | 1 |
| Florida Museum of Natural History (FLMNH/UF) | Gainesville, FL | USA | Preserved | 1 |

Note:
$N$ = number of specimens. Preserved samples were mostly in 4–5% formalin, but sometimes in 70% ethanol. Acronyms of other institutions cited, mostly in the species descriptions are as follows: CAS/CASIZ, California Academy of Sciences, Invertebrate Zoology, USA; MCZ, Museum of Comparative Zoology, Harvard, USA; NHM, The Natural History Museum, United Kingdom; UNIPD, Museum of Adriatic Zoology Giuseppe Olivi, Italy; UNIS_SCY, Laboratory of Zoology and Marine Biology in the University of Salento, Italy.

direct comparison with the species' redescription in *Scorrano et al. (2016)*. Specimen examinations included scaled photographs and measurements of features that involved "depth" or "thickness", which could not be acquired later from photographs. When necessary, a stereomicroscope was also used for observations. Morphological measurements were acquired from scaled photographs with the program Fiji (*Schindelin et al., 2012*).

Characteristics observed from medusae mainly followed *Dawson (2003)*, which included 24 characters, comprising continuous, meristic, and categorical features (Fig. 1, f1-30; Table S3). Sixteen extra characters were added (Fig. 1, f31-46; Table S3), either novel or from previous studies (*Gershwin, 2001*; *Chiaverano, Bayha & Graham, 2016*), mainly to unambiguously characterize categorical features, after observing their variation.

## Morphological data analyses

To account for differences in shape, morphology must be characterized regardless of size. As continuous and meristic features in our dataset may vary with size (*i.e.*, bell diameter—f1), we scaled all individuals to the same f1 by adapting the method of *Lleonart, Salat & Torres, (2000)*, which has been previously used for various animals including jellyfish (*Chiaverano, Bayha & Graham, 2016*). This method considers potential allometric differences that can occur between species or even within species across geographic localities. Specimens analyzed were therefore separated into geographic localities defined by countries, and usually also by region within the country (*e.g.*, southeast–SE). Size corrections followed the formula $Y^* = Y_i \, (f1_m/f1_i)^b$, in which the desired size-corrected feature ($Y^*$) equals its measurement in a specimen ($Y_i$) times the ratio between the average bell diameter in the locality group ($f1_m$) and the bell diameter of the specimen ($f1_i$), this raised to the power of the slope of the relationship between both log-transformed

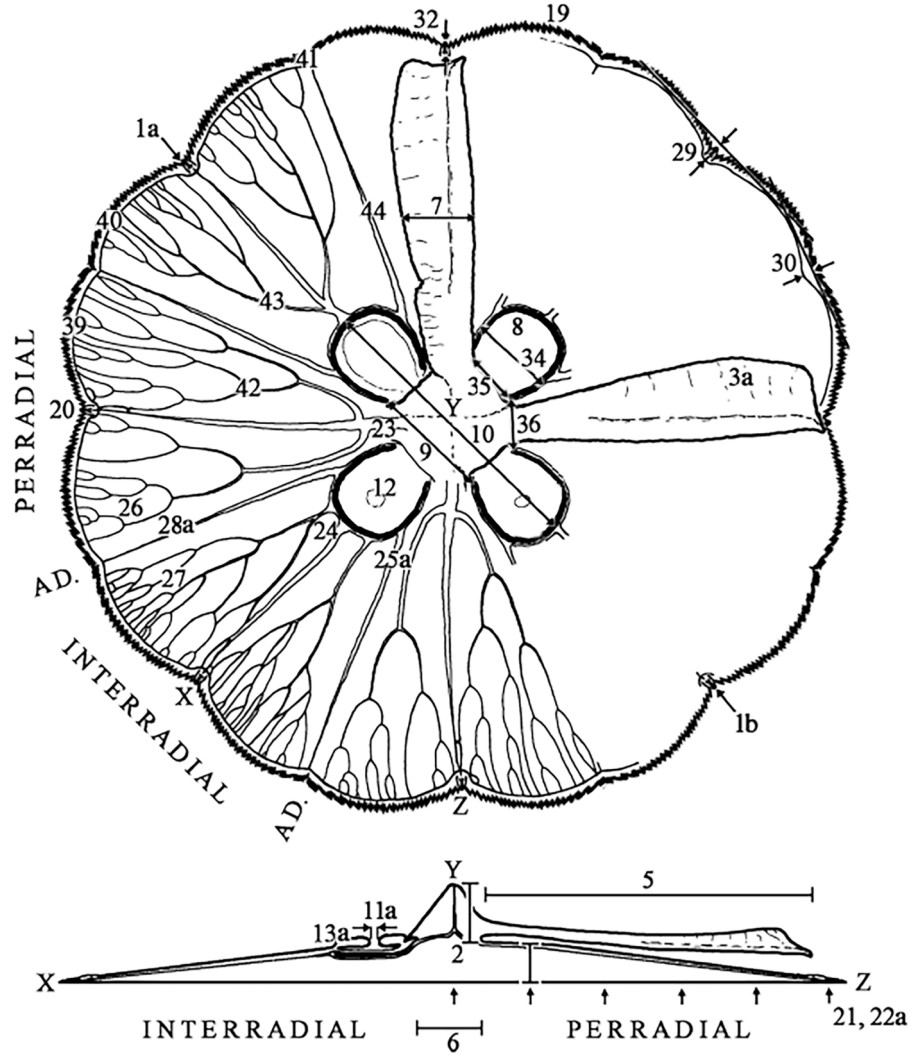

**Figure 1 Subumbrellar (top) and cross-sectional (bottom) views of an *Aurelia* medusa, illustrating most of the morphological features measured in this study.** Characters from *Dawson (2003)* span from 1–30 (with modifications indicated by an "a", except for 1a), while novel or from previous studies appear from 31–46. For more details and all features measured see Table S3. (Modified from *Scorrano et al., 2016*).

variables $Y$ and f1 from the entire dataset (*b*), as we did not have enough samples per locality group to obtain significant relationships.

Some morphological characters in 104 of the observed specimens were damaged, missing, or could not be measured by photographs. In this case, prior to the size correction mentioned above, we adapted Lleonart's method to perform estimations of these missing data, using the same formula as before, but considering $Y_i$ as the missing variable to be estimated and $Y^*$ as the average of the given variable in the locality group. When data for the variable were not present within the locality group, we used data from the closest locality, also accounting for morphological similarity when possible (see details in Table S1).

Features that were mostly invariable or that lacked a significant relationship with f1 were removed from further analyses, as they could bias the resulting dataset. Lastly, we standardized all variables to scale to a minimum of 0 and maximum of 1. Categorical features were excluded from analyses, as they may not be reliable due to the ambiguity seen in the specimens observed. Number of lobes (f19) and number of rhopalia (f20) were also removed as not to bias the results due to potential asymmetric development that may occur in some specimens. Medusae were not differentiated based on presence or absence of gonads, and consequently not between male and female, as size is the main factor influencing variation of morphological features.

In order to compare observed specimens based on the size corrected and scaled continuous and meristic morphological characters described above, we performed multidimensional scaling (MDS, Gower distance), with weighted average scores of variable contributions also mapped within. These analyses were separated into two sets, one that included all specimens with estimated missing data and subsequent size correction and scaling of variables, and another that excluded specimens with missing data, and therefore only size corrections and scaling were performed, to check for potential biases in estimations. Regarding these analysis sets we also computed a Mantel test with 9,999 permutations, to investigate the correlation between a geographic distance matrix (Euclidean distance, based on coordinates) and a morphological difference matrix (Gower distance, based on size corrected and scaled features), excluding aquarium specimens. The comparison of morphological measurements of *A. coerulea* from lab cultures and the species' redescription was performed by Welch's t-test. All of the corrections, estimations and further analyses mentioned above were performed using the software R version 4 (*R Core Team, 2020*) and relevant codes are available in GitHub (github.com/lawleyjw/Aurelia).

## Molecular data collection

Total DNA was extracted from oral arms of medusae, entire polyps, or entire ephyrae from specimens collected in the field or cultured in the laboratory at the University of São Paulo, using a protocol based on ammonium acetate, adapted from *Fetzner (1999)* (see Table S4 for details on samples used for molecular genetic analyses). From the mitochondrial genome, we amplified and sequenced two markers: a ~650-bp fragment of the large ribosomal RNA subunit (16S) and a ~650-bp fragment of the cytochrome c oxidase subunit I (COI) (primers derived from *Lawley et al., 2016*). From the nuclear genome, we obtained the internal transcribed spacer 1 (ITS1) with ~300-bp in length (primers jfITS1-5F, from *Dawson & Jacobs, 2001*; and ITS-R-28S-15, from *Cunha, Genzano & Marques, 2015*), and a ~650-bp fragment from the large ribosomal RNA subunit (28S) (primers Aa_L28S_260 and Aa_H28S_1078 from *Bayha et al., 2010*). Polymerase Chain Reaction (PCR) protocols followed standard procedures. Thermocycler profiles were conducted with initialization at 95 °C for three min, followed by 36–40 cycles of denaturation at 95 °C for 30 s, annealing at 46–58 °C (16S - 46 °C; COI - 52 °C; ITS1 - 57 °C; 28S - 58 °C) for 30–45 s, and extension at 72 °C for 1–2 min. Final extension was further conducted at 72 °C for 10 min. PCR products were purified using

Agencourt AMPure XP DNA Purification and Cleanup kit (Beckman Coulter Inc., Brea, CA, USA) and subsequently cycle-sequenced, with the same primers as before, to add fluorescently labeled dideoxy terminators. The above procedures were conducted at the Laboratory of Molecular Evolution (University of São Paulo) and chromatograms were generated on an Applied Biosystems 3730xl DNA Analyzer at the Laboratory of Plant Regulatory Network Signaling (University of São Paulo).

## Molecular analyses, species delimitation and descriptions

Sequenced chromatograms were assembled, trimmed and aligned in Geneious Prime 2019.0.4 (*Kearse et al., 2012*), which also included most sequences available in GenBank for *Aurelia* and some for *Drymonema dalmatinum* Haeckel, 1880 (Table S4), the chosen outgroup taxa as it was the most closely related Semaeostomeae to *Aurelia* (*Bayha et al., 2010*) with sequences for all markers herein studied. Alignments were performed using the software's implementation of MAFFT (*Katoh & Standley, 2013*), with the G-INS-i option and other default parameters, later visualized and edited manually to remove leading and trailing gap regions from variations in sequence length. Because the COI alignment did not present any alternative indel positions, which is common for protein-coding regions if introns are absent, the static alignment (*sensu Wheeler, 2001*) generated with MAFFT was used for phylogenetic analyses. This alignment was submitted to TNT ver 1.5 (*Goloboff & Catalano, 2016*) and analyzed under parsimony as the optimality criterion, using its New Technology searches (*Goloboff, 1999*; *Nixon, 1999*) with the following parameters: consense 10, css, rss, xss, rep 10, ratchet 50, drift 50, fuse 10. Node support was assessed by Goodman-Bremer values (*Goodman et al., 1982*; *Bremer, 1994*; *Grant & Kluge, 2008*), calculated by running a modified version of the script BREMER.RUN distributed with TNT, which considered 1,000 replicates with 10 repetitions of ratchet and drift (*Goloboff, 1999*; *Nixon, 1999*) in constrained searches. Bootstrap resampling frequencies were also calculated for nodes from 1,000 pseudoreplicates.

Considering ribosomal RNA regions, they commonly present insertions and deletions, which makes multiple sequence alignment more challenging (*Nagy et al., 2012*). To account for this, we submitted the resulting sequences of 16S, ITS1 and 28S to phylogenetic inference by direct optimization (*Wheeler, 1996*) using POY ver. 5.1.2 (*Wheeler et al., 2014*), under the parsimony optimality criterion. Tree search was performed by three independent one, three and six hour searches assuming equal rates for character transformations. All unique trees compiled from the above searches were submitted to tree refinement by the tree-fusing algorithm (*Goloboff, 1999*) and re-diagnosed with the iterative pass algorithm (*Wheeler, 2003a*). The resulting implied alignment (*sensu Wheeler, 2003b*) was submitted to TNT to verify the results, under the same parameters as described before, including the Goodman-Bremer support and bootstrap frequencies. The analyses run with POY were conducted in an IBM x3850 X5 server with eight processors Intel Xeon CPU E7-8870 2.40 GHz, housed at the Genetics and Evolutionary Biology Department (University of São Paulo).
Primary species hypotheses were considered based on previous mentions in the literature and when, in at least one single-marker phylogeny, species clades were monophyletic, had support ≥2 and resampling frequency ≥75. Then, markers were combined for a concatenated phylogenetic analysis. We imported the 16S, ITS1 and 28S implied alignments, and the COI static alignment, to Sequence Matrix ver 1.8 (*Vaidya, Lohman & Meier, 2011*), and selected mostly 3-4 sequences, when available, of each marker for each of the hypothesized species (see details of sequence composition in Table S5). Whenever possible, sequences of different markers within species were selected from the same specimen, but alternatively these sequence sets (terminal taxa) were chimeric, combined either from other specimens of the same locality, or the closest locality (Table S5). Sequence sets within each species hypothesis were selected from geographic regions as diverse as possible. The resulting file with combined alignments was analyzed in TNT as described previously. We also performed this concatenated phylogenetic analysis in IQ-TREE ver 1.6.12 (*Nguyen et al., 2015*) under maximum likelihood as the optimality criterion, using ModelFinder (*Kalyaanamoorthy et al., 2017*) for model selection and measuring bootstrap resampling frequencies and SH-aLRT (*Guindon et al., 2010*) from 1,000 pseudoreplicates. Alignments and trees retrieved in all molecular genetic analyses were deposited in Figshare (*Lawley et al., 2021*). All relevant codes used for genetic analyses are available in GitHub (github.com/lawleyjw/Aurelia).

For species delimitation (*sensu DeSalle & Goldstein, 2019*), primary species hypotheses were reassessed based on criteria from two lines of evidence: (1) species' monophyly and the clade's support (≥2), resampling frequency (≥75) and branch length (*ad hoc*) on the concatenated phylogeny; (2) species' distribution, based on collection localities of sequenced specimens (*ad hoc*). However, there are some caveats to this procedure. We recognize that phylogenetic analyses impose a hierarchy even on entities related tokogenetically (*Davis & Nixon, 1992*; *Grant et al., 2006*), and consequently species, which we herein consider as historical individuals, do not necessarily need to form a clade (*Kluge, 1990*; *Frost & Kluge, 1994*; *Skinner, 2004*). Therefore, branch lengths of the species' clades were also considered, as these are a measure of their differentiation. Nevertheless, due to variation in evolutionary rates and collection efforts, branch lengths may vary even across congeners (*Grant et al., 2006*). Considering species distributions can also be misleading, as there are likely multiple introductions in different *Aurelia* species (*Dawson, Gupta & England, 2005*), as well as sympatry (*Chiaverano, Bayha & Graham, 2016*). In spite of these caveats, these are clear and falsifiable criteria that can facilitate species discovery (*sensu DeSalle & Goldstein, 2019*) and diagnosability (*Frost et al., 1998*; *Grant et al., 2006*).

After species delimitation, diagnostic characters (*sensu ICZN (International Commission on Zoological Nomenclature), 1999*) were identified for each marker using the program YBYRÁ (*Machado, 2015*), considering the alignments and phylogenies of both single-marker and concatenated analyses. Reported diagnostic character-states for positions in the alignment are color-coded in the program's output, based on optimization of synapomorphies (*sensu Grant & Kluge, 2004*): white are ambiguous, and other colors are

unambiguous; black are unique and non-homoplastic; red are unique and homoplastic; and blue are non-unique and homoplastic (see further details in *Machado, 2015*).

We also calculated uncorrected pairwise distances (number of base mismatches divided by total sequence length, also known as uncorrected *p*), which were retrieved from the software Geneious. We did not use this measure to delimit species, as (1) pairwise distances only discriminate among samples, and therefore cannot diagnose any particular entity (*Frost, 2000*); (2) they fail to explain observed variation, as they cannot distinguish between symplesiomorphy and synapomorphy; and (3) due to variation in evolutionary rates that could occur even among congeners, as previously mentioned, there seems to be no justification to set an arbitrary distance as threshold for granting species status (*Grant et al., 2006*). Nevertheless, we evaluated the use of this measure across molecular genetic markers, as it can provide a rapid heuristic for species identification (*sensu DeSalle & Goldstein, 2019*) without the need of a complete phylogenetic analysis, in a similar way as dichotomous keys can be useful identification tools (*Grant, 2002*; *Grant et al., 2006*).

The electronic version of this article in Portable Document Format (PDF) will represent a published work according to the ICZN, and hence the new names contained in the electronic version are effectively published under that Code from the electronic edition alone. This published work and the nomenclatural acts it contains have been registered in ZooBank, the online registration system for the ICZN. The ZooBank LSIDs (Life Science Identifiers) can be resolved and the associated information viewed through any standard web browser by appending the LSID to the prefix http://zoobank.org/. The LSID for this publication is: http://zoobank.org/9CCDC703-92EB-4EDD-AB8F-F353941FEA1B. The online version of this work is archived and available from the following digital repositories: PeerJ, PubMed Central and CLOCKSS.

# RESULTS

## Morphological assessment

Before analyses were performed, morphological variation could already be observed among specimens of similar size from the same collection lot, as illustrated in Fig. 2. It is possible to see variation, for example, in the size of gastric pouches (f31) and sub-genital pores (f11a), as well as in the number of oral arm folds (f3a).

This variation was also observed after MDS analyses (with and without estimation of missing data), as neither presented a clear geographic structure on morphological dissimilarities (Figs. 3–4; correspondence of regions in Figs. 3–4 with specimen vouchers are shown in Figs. S1–S2). Specimens from lot YPM29380 from the northeastern coast of the USA for example, shown in Fig. 2, were sometimes more similar to specimens from very distant localities, such as the Maldives or the Marshall Islands, than to others in the same lot (highlighted in blue in Figs. 3–4). Another example of morphological resemblance among distant localities are the specimens from the northeastern coast of Canada and southwestern coast of the USA (highlighted in green in Figs. 3–4; Figs. 5A–5D). It is possible to see the similarity in the oral arms (f5 and f7), the size of the sub-genital pores (f11a) and in the branching pattern of interradial canals (f43) (Figs. 5A–5D). In that sense, even though some individuals within a locality or lot may

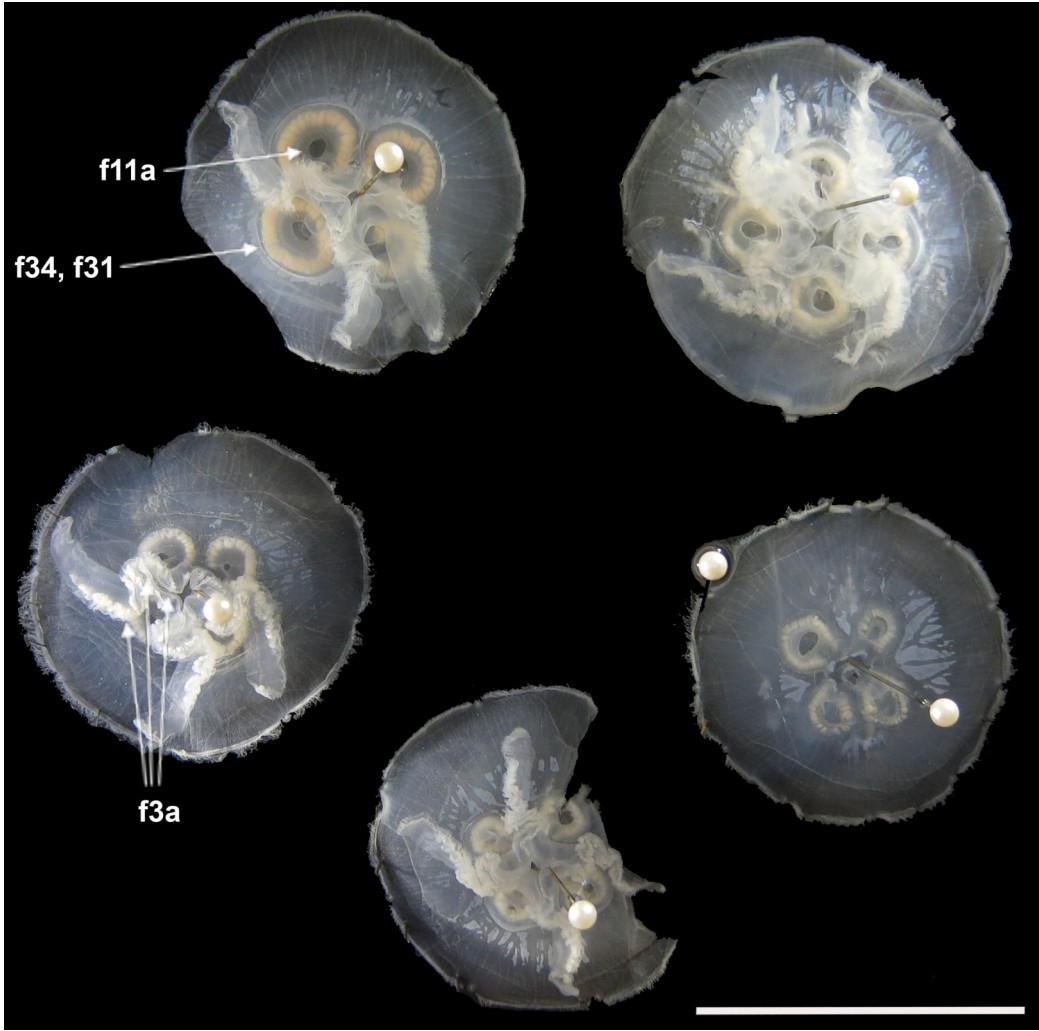

**Figure 2** *Aurelia* **medusae in lot YPM 29380, from Massachusetts, USA (highlighted in blue in Figs. 3–4, S1–S2).** f11a = Lateral sub-genital pore diameter; f34 = Lateral gastric diameter (furthest points); f31 = Size of gastric pouches; f3a = Number of oral arm folds (curving points per arm). For more details, see Fig. 1 and Tables S1–S3. Scale = 5 cm.

appear closer in the MDS, morphological variation within these groups still seem to be variable enough to overlap across very distant localities (Figs. 3–4). This is further emphasized by the spread of specimens that were identified from genetic sequences as *Aurelia coerulea* (presented further; highlighted in orange in Figs. 3–4). The only example in which many specimens from the same locality group are clustered separately, is in the case of *A. coerulea* individuals analyzed from the aquarium at Discovery Place, USA (circled in orange in Figs. 3–4). The only other specimens observed that could be identified from genetic sequences were three individuals from southeastern Brazil, two *Aurelia cebimarensis* sp. nov. (presented further; highlighted in pink in Figs. 3–4), and one *Aurelia mianzani* sp. nov. (presented further; highlighted in red in Fig. 3).

Even though there is no apparent geographic structure in dissimilarities, some of the measured characters seem to be more variable across all analyzed specimens, which are

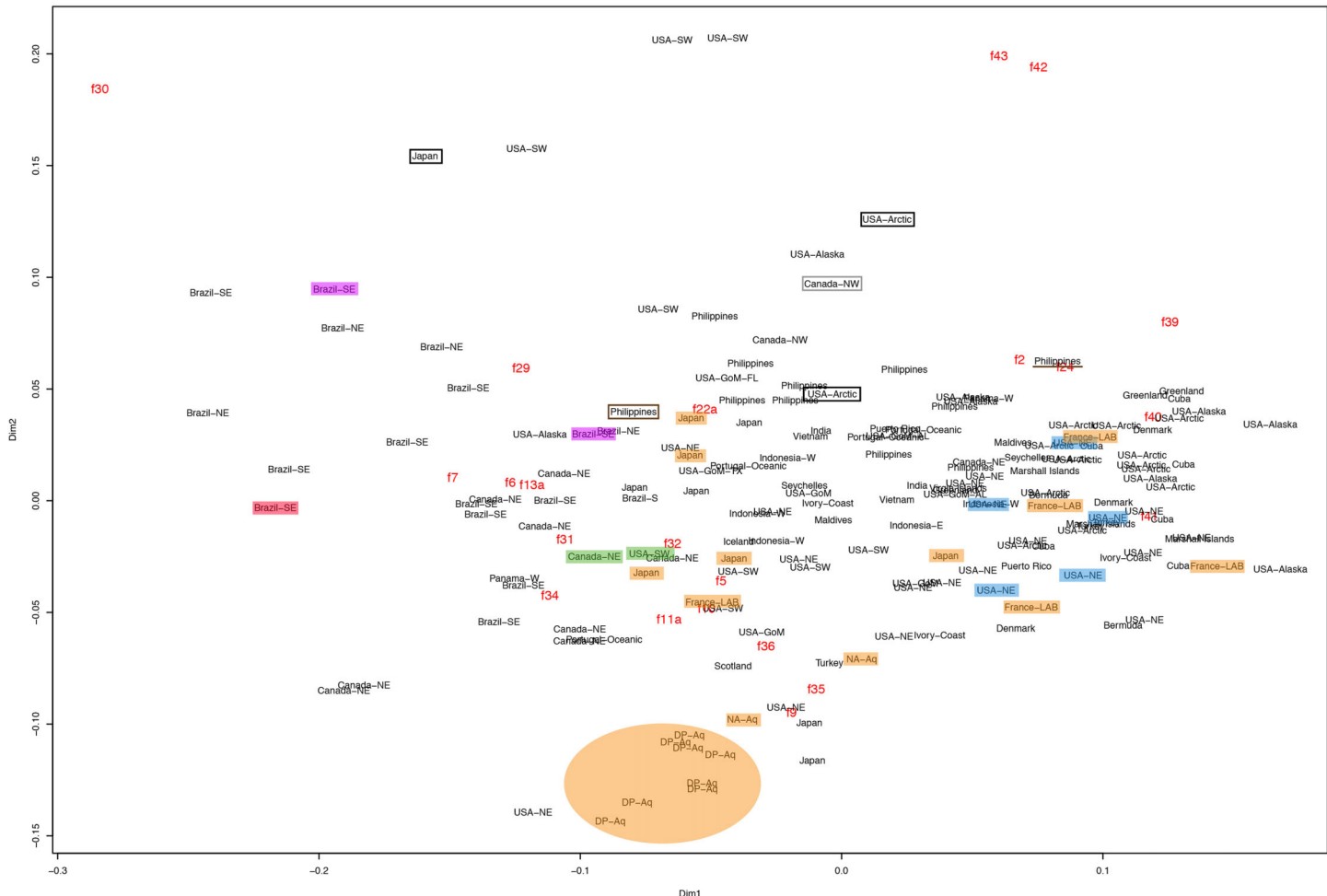

**Figure 3 Multidimensional scaling (MDS) of morphological features *with* estimation of missing data.** Specimens are depicted in black, as geographic locality or institution, and features appear in red, as weighted averages of their contributions. Specimens highlighted in blue from the northeastern USA appear in Fig. 2 (YPM29380) and those highlighted in green from northeastern Canada and the southwestern USA in Fig. 5 (USNM30988 and USNM92912-5, respectively). One of the specimens highlighted in orange from the aquarium at Discovery Place, USA, appears in Fig. 6A (DP3-4). One of the specimens in the black boxes from the Arctic appears in Fig. 6C (USNM 44243-2) and the specimen from northwestern Canada in the grey box appears in Fig. 6D (USNM92913-1). Specimens highlighted in orange are *Aurelia coerulea*, in pink are *Aurelia cebimarensis* sp. nov. and in red *Aurelia mianzani* sp. nov., identified based on genetic sequences (Table S4). See Fig. S1 for the exact correspondence to specimen vouchers, Table 1 for institution acronyms and Fig. 1, Tables S1–S3 for more information on specimens measured and morphological features.

represented closer to the edges of the morphological scape in the MDS (Figs. 3–4, characters in red). Specimens represented closer to these characters do not necessarily have the greatest values for it, but that character is the one that most contributed for the specimen's position in the MDS. Individuals from the aquarium at Discovery Place (circled in orange in Figs. 3–4) for example, seem to have the greatest distance between proximal edges of opposing gastric pouches, as well as between proximal tips in each gastric pouch (f9 and f35, respectively; Fig. 6A; Tables S1–S2). Rhopaliar and non-rhopaliar indentations (f29 and f30, respectively) are the largest in some specimens from Brazil (Fig. 6B) and one from the Philippines (brown square in Figs. 3–4; Tables S1–S2). Some specimens from the Arctic (Fig. 6C) and a specimen from Japan have
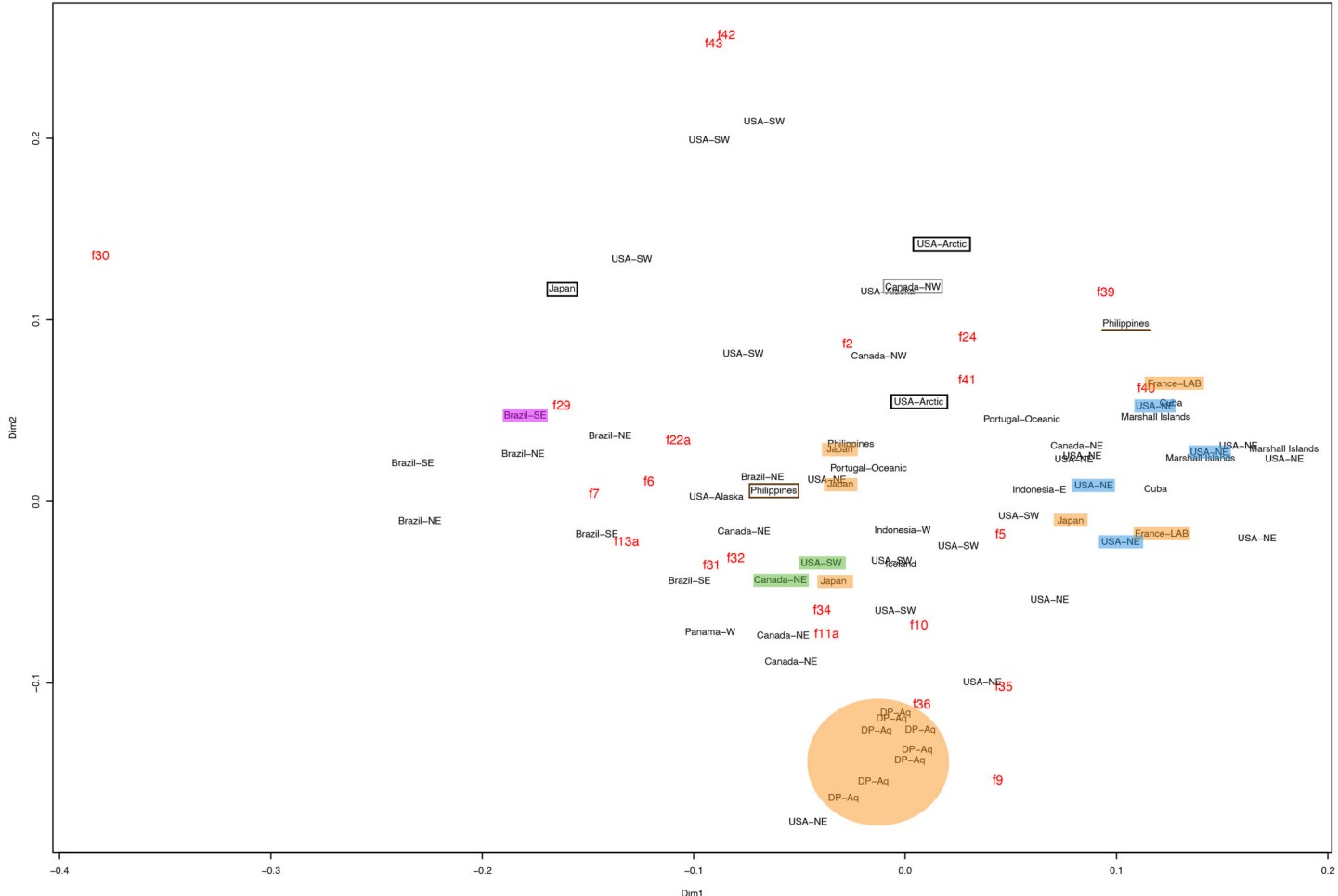

**Figure 4 Multidimensional scaling (MDS) of morphological features *without* estimation of missing data.** Specimens are depicted in black, as geographic locality or institution, and features appear in red, as weighted averages of their contributions. Specimens highlighted in blue from the northeastern USA appear in Fig. 2 (YPM29380) and those highlighted in green from northeastern Canada and the southwestern USA in Fig. 5 (USNM30988 and USNM92912-5, respectively). One of the specimens highlighted in orange from the aquarium at Discovery Place, USA, appears in Fig. 6A (DP3-4). One of the specimens in the black boxes from the Arctic appears in Fig. 6C (USNM 44243-2) and the specimen from northwestern Canada in the grey box appears in Fig. 6D (USNM92913-1). Specimens highlighted in orange are *Aurelia coerulea* and in pink *Aurelia cebimarensis* sp. nov., identified based on genetic sequences (Table S4). See Fig. S2 for the exact correspondence to specimen vouchers, Table 1 for institution acronyms and Fig. 1, Tables S1–S3 for more information on specimens measured and morphological features.

the highest number of perradial and interradial branching points (f42 and f43, respectively; black squares in Figs. 3–4), although it is also high in some specimens from the southwestern coast of the USA, and seems to contribute greatly for their position (Figs. 3–4; Tables S1–S2). In the same way, a specimen from northwestern Canada has the highest number of interradial terminations (f40; grey square in Figs. 3–4; Fig. 6D), although it is also high in a specimen from Cuba, contributing greatly to its position (Figs. 3–4; Tables S1–S2). The number of perradial terminations (f39), on the other hand, is higher in a specimen from the Philippines (underlined in brown in Figs. 3–4; Tables S1–S2).

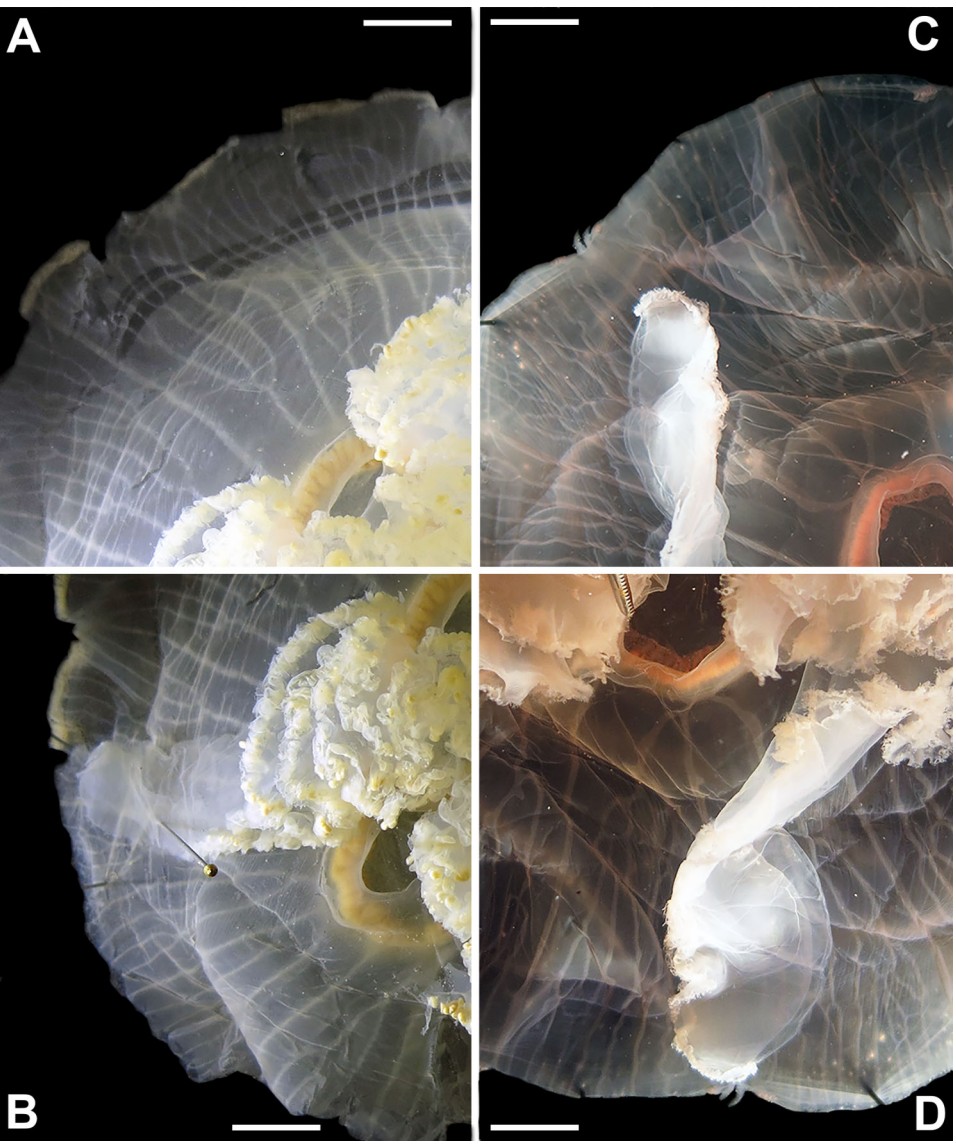

**Figure 5** *Aurelia* **medusae from northeastern Canada (USNM 30988) (A–B) and the southwestern USA (USNM 92912-5) (C–D) (highlighted in green in Figs. 3–4, S1–S2).** The images show an inter-radial sector, from the gastric pouch to the bell margin, and emphasize similarities on the oral arms (f5 and f7), the size of the sub-genital pores (f11a) and the branching pattern of radial canals (f43). For more details, see Fig. 1 and Tables S1–S3. Scales = 1 cm.

If specimens from neighboring localities had distinguishable morphological features, it could be argued that morphotypes could be identified regionally. The Mantel test revealed a different pattern, with a weak positive correlation in which specimens from nearby localities are more similar to each other compared to more distant localities, although only significant for the dataset with estimation of missing data ($R^2 = 0.067$, $p < 0.05$; without estimation of missing data, $R^2 = 0.083$, $p > 0.05$).

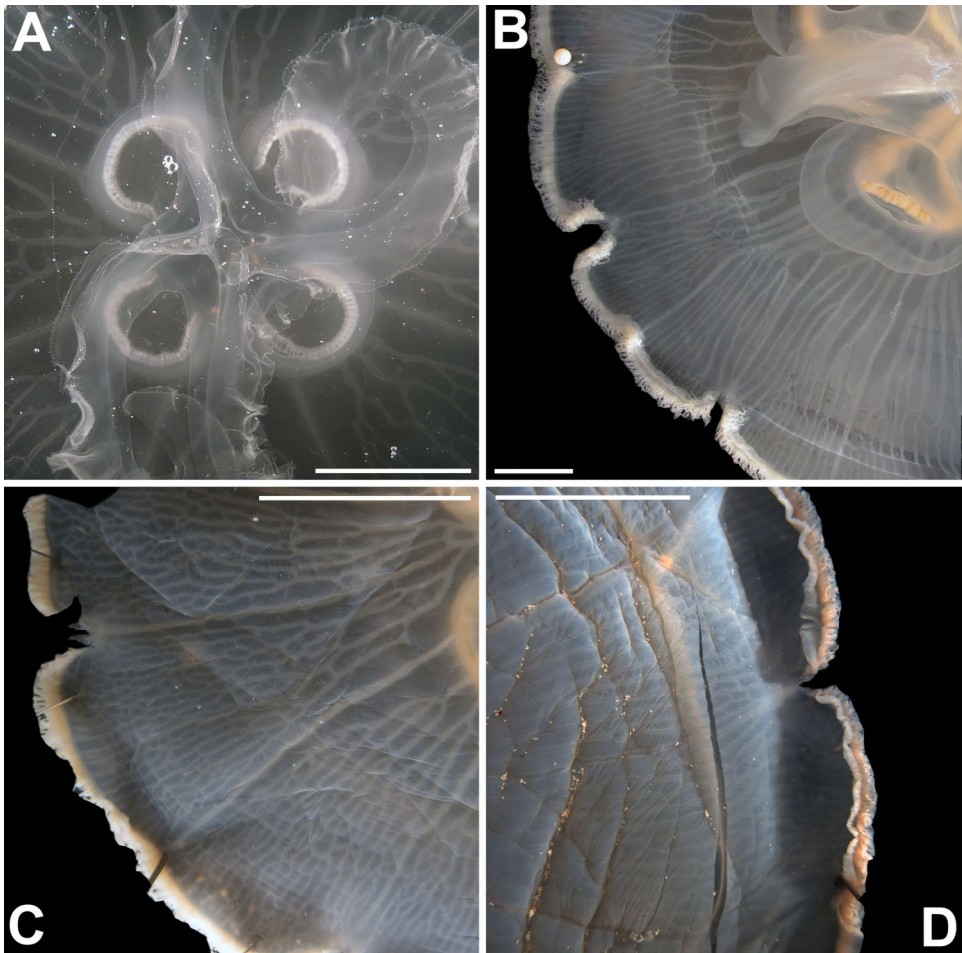

**Figure 6** *Aurelia* **from the aquarium at Discovery Place, USA (DP3-4) (A), Brazil (LAB08) (B), the Arctic (USNM 44243-2) (C) and northwestern Canada (USNM 92913-1) (D) (some of these are highlighted in Figs. 3–4, S1–S2).** The images illustrate some morphological features that are distinguished in these specimens, such as distance between proximal edges of opposing gastric pouches (f9) and between proximal tips in each gastric pouch (f35) (A), rhopaliar (f29) and non-rhopaliar indentations (f30) (B), number of perradial (f42) and interradial branching points (f43) (C), and number of interradial terminations (f40) (D). For more details, see Fig. 1 and Tables S1–S3. Scales = 2 cm.

## Morphological variability and diagnosis in *Aurelia coerulea*

In view of the high morphological variability observed, we compared the diagnostic features presented in the *A. coerulea* redescription, based on seven to 10 Mediterranean specimens (for more details see Table 2 in *Scorrano et al., 2016*), with two medusae from lab cultures (polyps originally from the North Sea) that were identified from genetic sequences as *A. coerulea* (presented further). By comparing Figs. 7A, 7C, some morphological differences can already be perceived, and this is further emphasized in Fig. 8, which highlights the significant differences in the continuous and meristic characters measured. Also, specimens cultured in the lab had a rounded hood covering the rhopalia, while Mediterranean specimens presented a triangular hood (Figs. 7B, 7D).

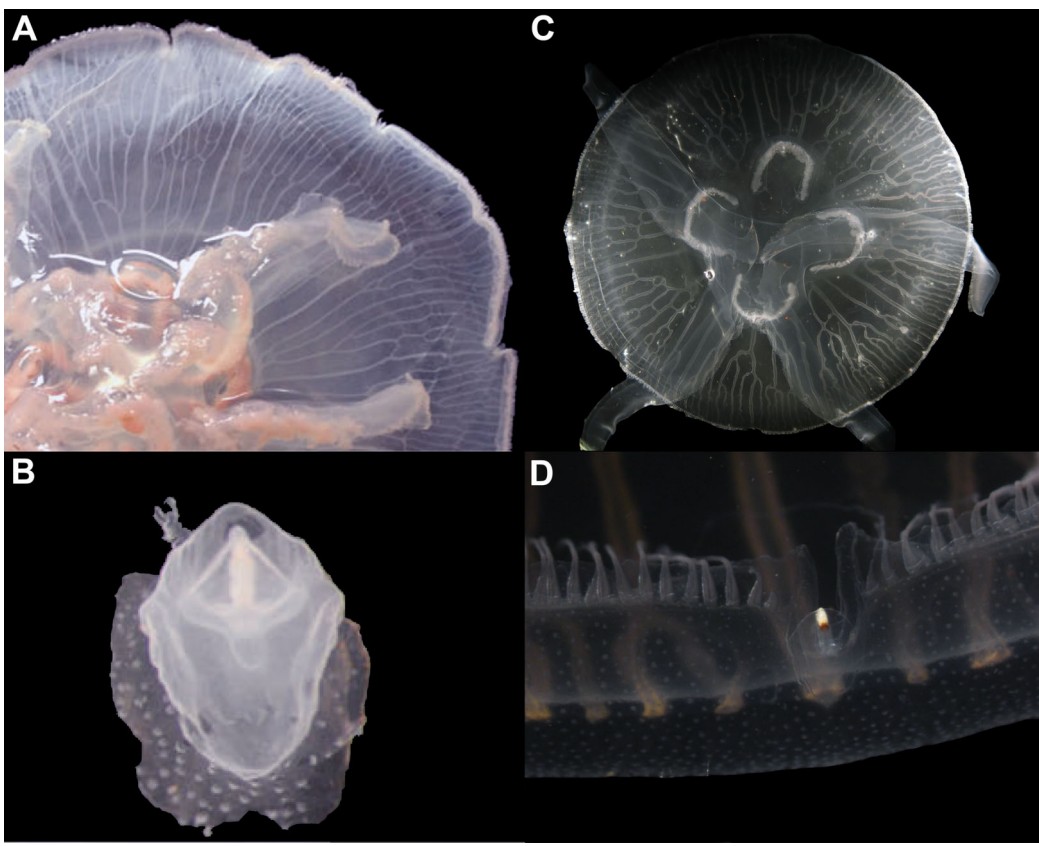

**Figure 7** *Aurelia coerulea* **from the Mediterranean (collected from the field) (A–B) and from lab cultures (originally from the North Sea) (C–D).** Some morphological differences can be perceived when comparing the medusa's overall appearance (A, C), such as the oral arms and gastric pouches. The hood that covers the rhopalia is also different, triangular in Mediterranean specimens (B) and rounded in cultured specimens (D). A–B, bell diameter (f1) = 12.5 cm (images A and B were adapted from *Scorrano et al., 2016*); C-D, f1 = 7.5 cm.

## Species delimitation

The concatenated phylogenetic analysis, which combined all markers herein studied (16S, COI, ITS1 and 28S), revealed 28 species hypotheses (Fig. 9), of which seven had already been described and are currently recognized as valid (*A. marginalis*, *A. solida*, *A. labiata*, *A. relicta*, *A. aurita*, *A. coerulea* and *A. limbata*; based on *Collins, Jarms & Morandini, 2020*). Of the 21 remaining, two were collected and sequenced in this study for the first time (*Aurelia ayla* sp. nov. and *Aurelia insularia* sp. nov.), and eight had been previously sequenced or even considered as species hypotheses but not formally described (*Aurelia miyakei* sp. nov., *Aurelia mianzani* sp. nov., *Aurelia rara* sp. nov., *Aurelia montyi* sp. nov., *Aurelia smithsoniana* sp. nov., *Aurelia cebimarensis* sp. nov., *Aurelia malayensis* sp. nov. and *Aurelia columbia* sp. nov.). Four species were resurrected (*Aurelia clausa* *Lesson, 1830*, *Aurelia dubia* *Vanhöffen, 1888*, *Aurelia persea* (*Forskål, 1775*) and *Aurelia hyalina* *Brandt, 1835*) and the other seven remain undescribed (sp. 3, 7, 12, 13, 14, 17 and 18), as their distributions did not match any available names, or there was either no type material available or they were already under description. A map

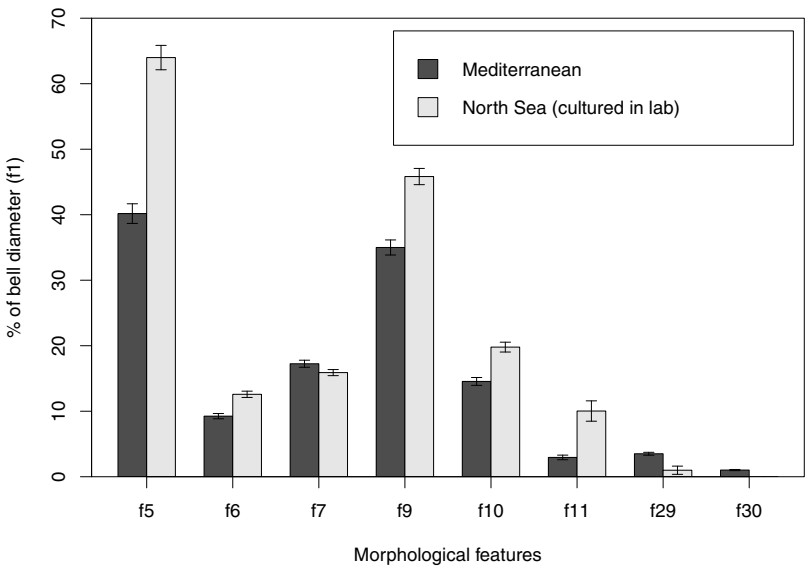

**Figure 8 Comparison of morphological features measured from seven to ten Mediterranean specimens (Table 2 from *Scorrano et al., 2016*) and from two lab-cultured medusae of *Aurelia coerulea*.** Averages and standard deviations are presented for each morphological feature, for which a Welch's t-test returned significant differences between Mediterranean and lab-cultured specimens for all features (*p* < 0.05). For references on morphological features, see Fig. 1 and Table S3.

of type localities (or sampling localities in the case of undescribed species) for *Aurelia* species hypotheses is shown in Fig. 10, while the distribution ranges are described in the 'Systematic account' section (for specific localities of sequenced specimens see Table S4).

In the concatenated phylogeny, counts of diagnostic genetic characters within each unambiguous synapomorphy category are represented for species clades (Fig. 9). Six of the 28 species hypothesized do not present unique and non-homoplastic synapomorphies (in black, Fig. 9), which does not seem to be related to clade support, resampling frequency or branch length. Nevertheless, there were synapomorphies present in at least one of the unambiguous categories for every species. Ten species had sequences of the four genetic markers used for at least one sequence set (terminal taxon) in the concatenated phylogenetic analysis (circles with grey or black coloring for every quarter in Fig. 9). Complete sequence sets (with sequences of all markers) represented 30% of terminal taxa (see completeness ratios in Table S5). Non-chimeric sequence sets (only containing sequences from the same specimen) represented 49% of terminal taxa, and within the chimeric sequence sets an average of 60% of sequences within a set were not from the same specimen (see chimerism ratios in Table S5).

A concatenated phylogenetic analysis was also performed using maximum likelihood as the optimality criterion, in which some relationships between species differ from the parsimony analysis but species hypotheses are the same (Fig. S3). Single-marker phylogenies, which were used to construct primary species hypothesis, are represented in Figs. S4–S7, and the diagnostic synapomorphies derived from these are included as counts for each species in the 'Systematic account' section (diagnostic positions for each

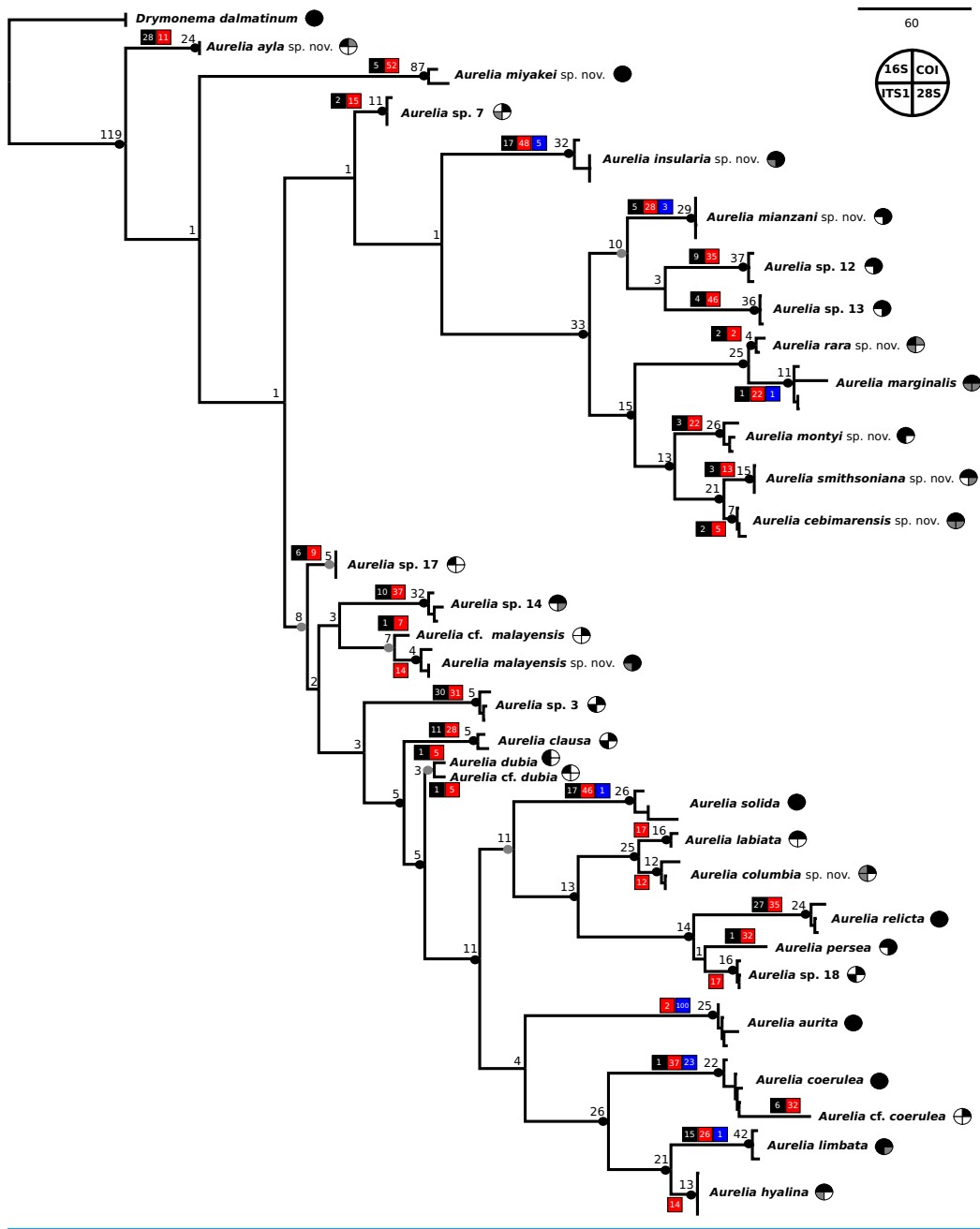

**Figure 9 Concatenated phylogenetic analysis, indicating relationships between 28 *Aurelia* species hypotheses.** This analysis combined markers 16S, COI, ITS1 and 28S, reconstructing relationships under parsimony as the optimality criterion. Unambiguous synapomorphies for each species are represented as counts of each retrieved category (black are unique and non-homoplastic; red are unique and homoplastic; and blue are non-unique and homoplastic; see further details in *Machado, 2015*). Circles next to species names indicate the presence or absence of a genetic marker (following legend) and its respective synapomorphies (black = marker and synapomorphies present, grey = marker present but synapomorphies absent, white = marker absent). Numbers on nodes indicate Goodman-Bremer support values and colored circles represent bootstrap resampling frequencies (black = ≥95, grey = ≥75, absent = <75). Scale bar represents the number of nucleotide transformations. Table S4 contains further details on sequences used to reconstruct this phylogeny. A concatenated phylogenetic analysis was also performed under maximum likelihood as the optimality criterion and is available in Fig. S3. Single-marker phylogenies are presented in Figs. S4–S7.
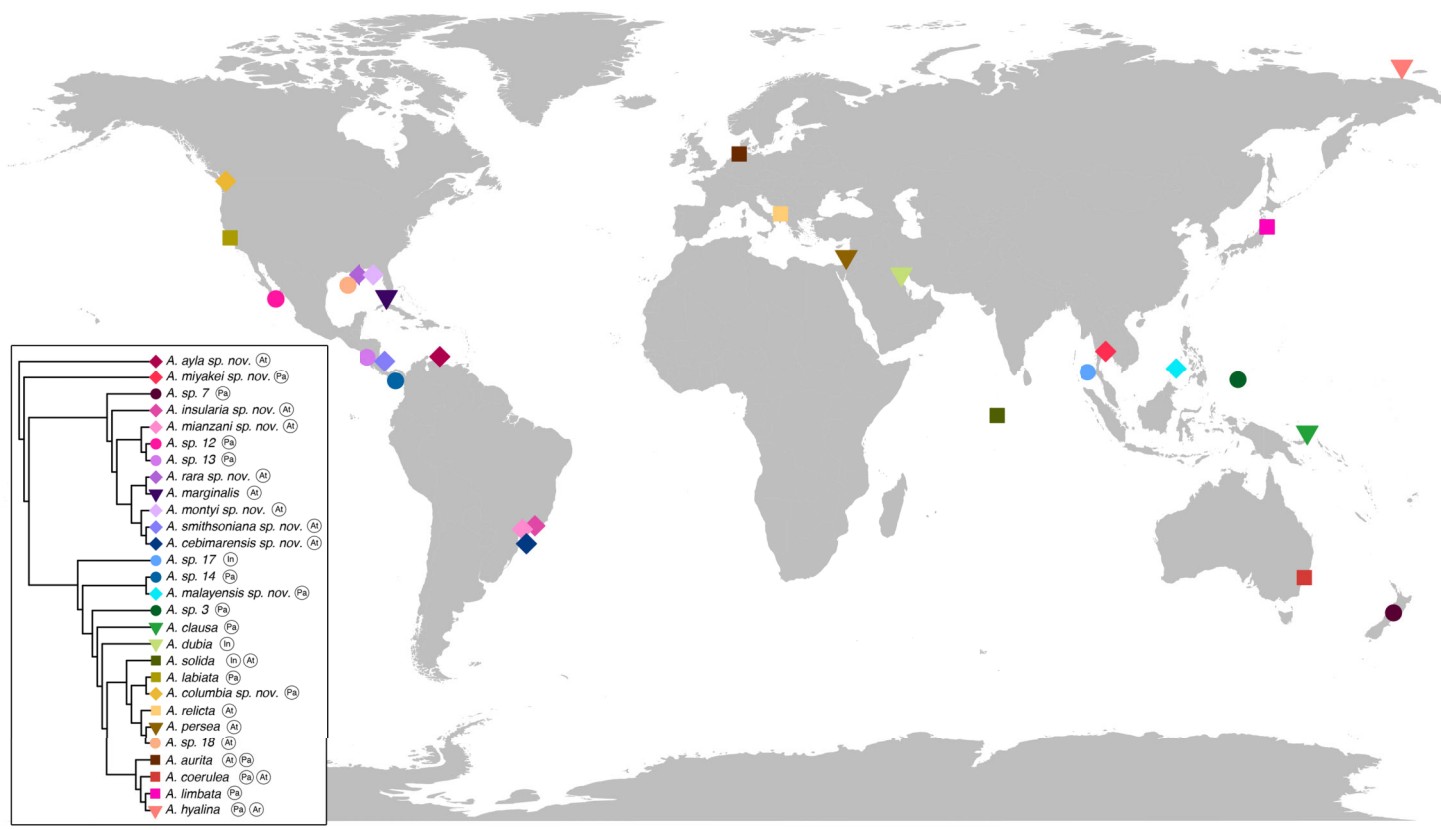

**Figure 10 Sampling or type localities for *Aurelia* species treated herein.** Symbol shapes correspond to existing (square), resurrected (inverted triangle), new (diamond) and undescribed (circle) species. Oceans where each species is distributed are indicated in circles next to species names in the inset concatenated phylogenetic tree (detailed in Fig. 9). Details on the distribution range of each species is described in the 'Systematic account' section and the precise localities of sequenced specimens are presented in Table S4.

species can be found in Table S6). Considering uncorrected pairwise distances, they are reported for single-marker alignments as frequency histograms in Fig. 11. There was an overlap between intra- and interspecific distances for all markers, although COI was the marker that presented an overall gap between intra- and interspecific distances (Fig. 11; for details on intra- and interspecific distances for each marker see Table S7). Nevertheless, ITS1 had the greatest number of unique and non-homoplastic synapomorphies when comparing single-marker analyses, while COI had the least amount (Table S6). As with synapomorphies in the concatenated phylogenetic analysis, not all species hypotheses recovered in single-marker analyses had unique and non-homoplastic synapomorphies (Table S6).

## Systematic account
**Phylum Cnidaria** *Verrill, 1865*
**Class Scyphozoa** *Goette, 1887*
**Order Semaeostomeae** *Agassiz, 1862*
**Family Ulmaridae** *Haeckel, 1880*
**Genus *Aurelia*** *Lamarck, 1816*

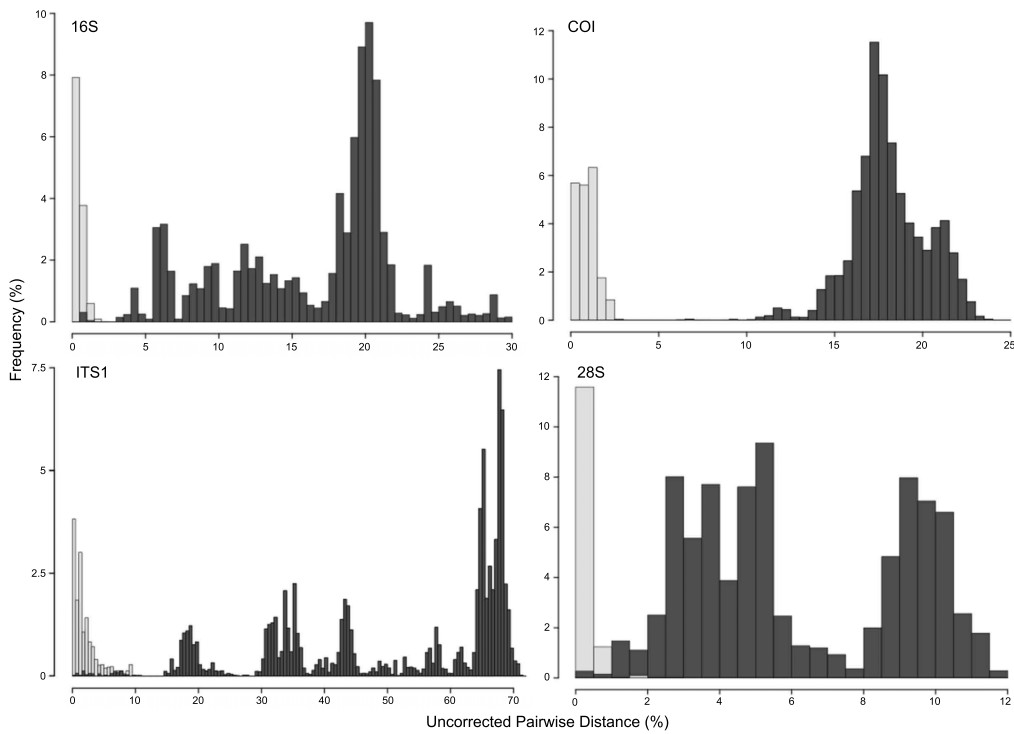

**Figure 11 Frequency histograms of uncorrected pairwise distances (%) from 16S, COI, ITS1 and 28S alignments.** Intraspecific distances are represented in light-grey, while interspecific distances appear in dark-grey (see details in Table S7).

**Diagnosis:** Ulmarid scyphomedusae with unbranched oral arms surrounding the mouth; interradial gastric pouches with much folded gonads (when present), unconnected from each other and with external subgenital pores, usually ranging in shape from a flat-u to a drop-shaped invagination; radial canals branching and sometimes anastomosing, extending outwards to margin from central stomach; ring canal present; lappet-like structures and numerous (commonly more than 1,000) small tentacles arising from exumbrella just above the margin; marginal rhopalia usually on the center of the perimeter of each radius, commonly resulting on a bell indentation (marginal cleft) that marks the division between velar lobes; non-rhopaliar indentations can be present and therefore increase divisions between velar lobes; usually presents tetramerous radial symmetry (compiled from *Mayer, 1910*; *Kramp, 1961*; *Russell, 1970*; *Calder, 2009*; *Scorrano et al., 2016*; *Jarms & Morandini, 2019*; and own observations).

**Type species:** *Aurelia aurita* (*Linnaeus, 1758*)

Species hypotheses are presented below in the order they appear in the concatenated phylogenetic analyses, from top to bottom (Fig. 9). For precise localities of sequenced specimens, see Table S4. Molecular genetic diagnosis, with details on synapomorphy categories and positions are given in Table S6. A brief morphological description is provided, when available, for all life cycle stages of species considered herein (Table S8).

*Aurelia ayla* Lawley, Gamero-Mora, Maronna, Chiaverano, Stampar, Hopcroft, Collins and Morandini **sp. nov.**

**Type material:** Holotype: Tissue (Medusa), USNM 1622161. Paratype: Tissue (Medusa), USNM 1622169.

**Type locality:** Oil slick leap, Kralendijk, Bonaire, the Netherlands (Fig. 10).

**Etymology:** Derived from the Turkish word *ayla*, meaning "halo of light around the moon", in honor of the daughter of AGC (co-author in this study), who shares the same name.

**Distribution:** Currently known only from the type locality (Table S4).

**Diagnosis:** There were 20 diagnostic positions for 16S and 20 for COI (Table S6).

**Remarks:** Interestingly, this species does not fall within the clade that includes most western Atlantic species. Further increasing the dataset with more molecular genetic markers and specimen collections, especially from the southeastern Atlantic and Indian Oceans, could resolve this matter, as it could not only be an effect of undersampling but also a case of introduction from another locality, which is not unprecedented in this genus (*Dawson, Gupta & England, 2005*). For a brief morphological description of the medusa stage see Table S8.

*Aurelia miyakei* Lawley, Gamero-Mora, Maronna, Chiaverano, Stampar, Hopcroft, Collins and Morandini **sp. nov.**

*Aurelia* MCA lineage *Schroth et al., 2002*. *Dawson, Gupta & England, 2005*. *Aurelia* sp. 11 *Dawson, Gupta & England, 2005*. *He et al., 2015*; *Chiaverano, Bayha & Graham, 2016*; *Dong, 2018*.

**Type material:** Holotype: Polyps, MZUSP 8654. Paratypes: Tissue (Polyps), MZUSP 8655.

**Type locality:** Gulf of Thailand, near Saen Suk, Thailand (Fig. 10).

**Etymology:** Named after Prof. Dr. Hiroshi Miyake (Kitasato University, Japan), for his prominent research on jellyfish and constant collaborative efforts, including providing polyps from this species.

**Distribution:** Gulf of Thailand and Kwajalein, Marshall Islands (Table S4).

**Diagnosis:** There were 17 diagnostic positions for 16S, 20 for COI, 20 for ITS1 and 17 for 28S (Table S6).

**Remarks:** Polyps were present in collected material from nearby the Institute of Marine Science, Burapha University, Saen Suk, Thailand. For a brief morphological description of the polyp stage see Table S8.

***Aurelia insularia*** Lawley, Gamero-Mora, Maronna, Chiaverano, Stampar, Hopcroft, Collins and Morandini **sp. nov.**

*Aurelia* sp. 2 *Gambill & Jarms, 2014*.

**Type material:** Holotype: Polyps, MZUSP 8648. Paratypes: Polyps, MZUSP 8647; Tissue (Polyps), MZUSP 8649.

**Type locality:** Pinguino Wreck, Ilha Grande, Rio de Janeiro, Brazil (Fig. 10).

**Etymology:** Derived from the Latin word *insularis*, meaning "of islands", in reference to the recorded occurrence of polyps mostly on or near islands.

**Distribution:** Mostly on or near islands in the south and southeastern coasts of Brazil, as well as on Key Largo, Florida, USA (Table S4).

**Diagnosis:** Polyps present 27–33 tentacles (Table S8; see remarks below). There were 20 diagnostic positions for 16S, 20 for COI, 20 for ITS1 and 11 for 28S (Table S6).

**Remarks:** Polyps of this species were first collected in Ilha Grande, Rio de Janeiro, in 2000, by Prof. Dr. Alvaro E. Migotto (CEBIMar-USP, Brazil). After *Dawson & Jacobs (2001)* identified sequences of medusae from the coast of São Paulo state as *Aurelia* sp. 2 (herein described as *Aurelia cebimarensis* sp. nov.), polyps from Ilha Grande were also assigned to this species. *Gambill & Jarms (2014)*, in their study of *Aurelia* scyphistomae and ephyrae, also recognized the polyps from Ilha Grande as *A*. sp. 2, and it was the only population that had 27-33 tentacles, while all others in the study presented ~16 tentacles. This remains as the only morphological diagnostic character for this species, even though morphological plasticity has also been reported in *Aurelia* polyps and ephyrae (*Chiaverano & Graham, 2017*). Nevertheless, there are unambiguous molecular genetic characters to support this species' diagnosis. More recently, this species was detected on Key Largo (Florida, USA) with the use of eDNA techniques (*Ames et al., 2021*). For a brief morphological description of the polyp and ephyra stages see Table S8.

***Aurelia mianzani*** Lawley, Gamero-Mora, Maronna, Chiaverano, Stampar, Hopcroft, Collins and Morandini **sp. nov.**

*Aurelia* sp. AA2501 South West Atlantic *Ramšak, Stopar & Malej, 2012*.
*Aurelia* sp. 16 *Gómez-Daglio & Dawson, 2017*. *Abboud, Gómez-Daglio & Dawson, 2018*; *Dong, 2018*.

**Type material:** Holotype: Medusa, MZUSP 8652. Paratype: Tissue (Medusa), MZUSP 8653.

**Type locality:** Praia do Segredo, São Paulo, Brazil (Fig. 10).

**Etymology:** In honor of Dr. Hermes W. Mianzan (INIDEP, Argentina), who collected some of the sequenced specimens of this species, and for his lifelong contributions and dedication to understanding jellyfish biology and ecology in the Southwestern Atlantic.

**Distribution:** Praia do Segredo, São Paulo, Brazil and Bahía Samborombón, Buenos Aires, Argentina (Table S4).

**Diagnosis:** There were 10 diagnostic positions for 16S, 19 for COI and five for 28S (Table S6).

**Remarks:** In *Ramšak, Stopar & Malej (2012)*, the specimen collected in the Southwestern Atlantic appeared as sister taxa to a specimen from the Mljet lakes, Croatia (currently known as *Aurelia relicta*), in their combined-marker phylogeny. In our single-marker phylogenies, we observed that the COI sequence from that study fell within a clade alongside the other sequences from Argentina (Fig. S5), while the ITS1 sequence from that same specimen fell within the *A. relicta* clade (Fig. S6). This could be explained by contamination in sequencing the ITS1, as in the aforementioned study, *A. relicta* specimens from the Mljet lakes were also being processed. As other sequences from Argentina were available, the specimen from *Ramšak, Stopar & Malej (2012)* was disregarded from our concatenated phylogenetic analysis.

Interestingly, this species forms a clade that is sister to the clade containing both *Aurelia* sp. 12 and *Aurelia* sp. 13 from the eastern Pacific, all nested within a clade shared by most western Atlantic species (Fig. 9). This diversification across the Isthmus of Panama has been reported for other cnidarians (*Stampar et al., 2012*; *Gómez-Daglio & Dawson, 2017*). Further biogeographical studies and increased sampling can verify this matter, as well as the curious position of the other eastern Pacific species *Aurelia* sp. 14 on the concatenated phylogeny (Fig. 9). For a brief morphological description of the medusa stage see Table S8.

**Aurelia rara** Lawley, Gamero-Mora, Maronna, Chiaverano, Stampar, Hopcroft, Collins and Morandini **sp. nov.**

*Aurelia* sp. DI'03-4 *Chiaverano, Bayha & Graham, 2016*.

**Type material:** Holotype: Tissue (Medusa), USNM 1643584. Paratype: Tissue (Medusa), USNM 1643585.

**Type locality:** Dauphin Island, Alabama, United States of America (Fig. 10).

**Etymology:** Derived from the Latin word *rarus*, meaning "rare" or "uncommon", due to its elusive occurrence among the other two species collected in the same locality (*Aurelia montyi* sp. nov. and *Aurelia marginalis*, herein considered).

**Distribution:** Currently known only from the type locality (Table S4).

**Diagnosis:** There were six diagnostic positions for 16S and 18 for COI (Table S6).

**Remarks:** *Chiaverano, Bayha & Graham (2016)* sequenced one specimen from Dauphin Island (DI'03-4), which in their COI phylogeny did not group with any other species, while in the ITS1 phylogeny fell within *A. marginalis* (previously recognized as *Aurelia* sp. 9; also see Table S4 and Figs. S5–S6). In our concatenated phylogeny (Fig. 9) however, as

for the 16S and COI single-marker phylogenies (Figs. S4–S5), this species fell in a separate clade from *A. marginalis*. These two taxa considered as separate species should not come as a surprise, as previous studies also demonstrated the occurrence of another sympatric species in the area, *A. montyi* sp. nov. (recognized as *Aurelia* cf. sp. 2 in *Chiaverano, Bayha & Graham, 2016*; described herein). For a brief morphological description of the medusa stage see Table S8.

**Aurelia marginalis** *Agassiz, 1862*

*Aurelia* sp. 9 *Dawson, Gupta & England, 2005*. *Ki et al., 2008*; *Ramšak, Stopar & Malej, 2012*; *Dong, Liu & Liu, 2015*; *He et al., 2015*; *Chiaverano, Bayha & Graham, 2016*; *Scorrano et al., 2016*; *Chiaverano & Graham, 2017*; *Gómez-Daglio & Dawson, 2017*; *Abboud, Gómez-Daglio & Dawson, 2018*; *Dong, 2018*, *Frolova & Miglietta, 2020*.

**Type material:** Holotype: Medusa, MCZ 352.

**Type locality:** Key West, Florida, United States of America (Fig. 10).

**Distribution:** Across the Gulf of Mexico (Table S4).

**Diagnosis:** There were 20 diagnostic positions for COI and three for 28S (Table S6).

**Remarks:** Specimens of both previously recognized *A.* sp. 9 (here synonymized) and *A.* cf. sp. 2 lineages (herein described as *A. montyi* sp. nov.) have been collected in the Florida Keys (Long Key, Florida, USA), which is within the "reefs of Florida", locality cited in the description of *Aurelia marginalis*, and very near the type specimen's locality. Even though we portray in this study the unreliability of morphological data for species recognition (further discussed), we decided to synonymize *A.* sp. 9 under *A. marginalis*, as in this species' description, *Agassiz (1862)* mentions the distinct rose color of the gonads, which is also presented by *Chiaverano, Bayha & Graham (2016)* for *A.* sp. 9 when compared to *A.* cf. sp. 2 (*A. montyi* sp. nov.) (see Fig. 1 in *Chiaverano, Bayha & Graham, 2016*). Nevertheless, this should not be used as diagnostic, as color has been previously reported on holding no value for systematics in this genus (*Kramp, 1968*), and even in other Medusozoa (*Lampert et al., 2011*; *Holst & Laakmann, 2014*).

 *A. marginalis* was resurrected by *Calder (2009)*, due to differences with specimens from the northeastern USA, which were reported as more similar to *Aurelia aurita* from northern Europe. These differences came mostly from polyps, on their free amino acid composition, nematocyst types, morphology, and asexual reproduction (*Calder, 2009*). The use of morphological characters in polyps to recognize different *Aurelia* species has been reported as problematic (*Gambill & Jarms, 2014*), which was further corroborated by the possibility of morphological plasticity due to environmental differences (*Chiaverano & Graham, 2017*). The use of nematocyst types for species recognition can also be problematic (*Francis, 2004*; *Acuña, Ricci & Excoffon, 2011*). Therefore, we do not report these as diagnostic for this species, but we corroborate the resurrection by *Calder (2009)* with a molecular genetic diagnosis. Other synonyms for this species have been presented, but we refrain from maintaining them, as they could belong to other species

present in the Gulf of Mexico, and until now we could not confirm it. For a brief morphological description of the polyp, ephyra and medusa stages see Table S8.

***Aurelia montyi*** Lawley, Gamero-Mora, Maronna, Chiaverano, Stampar, Hopcroft, Collins and Morandini **sp. nov.**

*Aurelia* cf. sp. 2 *Chiaverano, Bayha & Graham, 2016*.

**Type material:** Holotype: Tissue (Medusa), USNM 1643581. Paratypes: Tissue (Medusa), USNM 1643582-1643583.

**Type locality:** Dauphin Island, Alabama, United States of America (Fig. 10).

**Etymology:** Named after Dr. William "Monty" Graham (Florida Institute of Oceanography, USA), who was a pioneer in ecological studies with *Aurelia* in the Gulf of Mexico and former advisor of LMC (co-author in this study), both of which collected and sequenced most of the specimens that belong to this species.

**Distribution:** Eastern Gulf of Mexico (Table S4).

**Diagnosis:** There were five diagnostic positions for 16S, 11 for COI and six for ITS1 (Table S6).

**Remarks:** This species was considered as *A.* cf. sp. 2 because it was in the same clade as *A.* sp. 2 (herein described as *A. cebimarensis* sp. nov.) in the ITS1 phylogeny (see Fig. 5 in *Chiaverano, Bayha & Graham, 2016*), even though there were considerable branch lengths separating them and they were reciprocally monophyletic in the COI phylogeny (see Fig. 4 in *Chiaverano, Bayha & Graham, 2016*). Nevertheless, by including more genetic data, all of the phylogenies that included this species returned it as a separate clade (Fig. 9; Figs. S4–S6), which seems enough evidence now to corroborate this species' hypothesis. For a brief morphological description of the medusa stage see Table S8.

***Aurelia smithsoniana*** Lawley, Gamero-Mora, Maronna, Chiaverano, Stampar, Hopcroft, Collins and Morandini **sp. nov.**

*Aurelia* sp. 15 *Gómez-Daglio & Dawson, 2017*. *Abboud, Gómez-Daglio & Dawson, 2018*; *Dong, 2018*.

**Type material:** Holotype: DNA extraction, MZUSP 8656.

**Type locality:** Bocatorito Bay, Bocas del Toro, Panama (Fig. 10).

**Etymology:** Named after the Smithsonian Tropical Research Institute, in Bocas del Toro, Panama, which has supported studies in marine science for decades, especially in the Bocas del Toro area, where this species is distributed.

**Distribution:** Bocas del Toro, Panama (Table S4).

**Diagnosis:** There were 10 diagnostic positions for 16S and seven for COI (Table S6).

**Remarks:** In our 28S phylogeny (Fig. S7), this species appears in a single clade with *A. cebimarensis* sp. nov., although they appear reciprocally monophyletic in the 16S and COI phylogenies (Figs. S4–S5), and more importantly in the concatenated phylogeny (Fig. 9). Furthermore, even though there are reported cases of sympatric *Aurelia* species (*Chiaverano, Bayha & Graham, 2016*) and multiple introductions (*Dawson, Gupta & England, 2005*), the disjunct distribution of these sister species in neighboring but different biogeographic realms (*Costello et al., 2017*), as well as different large marine ecosystems (*Sherman, 1991*), could be further evidence of lineage separation. For a brief morphological description the medusa stage see Table S8.

*Aurelia cebimarensis* Lawley, Gamero-Mora, Maronna, Chiaverano, Stampar, Hopcroft, Collins and Morandini **sp. nov.**

*Aurelia* sp. 2 *Dawson & Jacobs, 2001*. *Dawson, 2003*; *Dawson, Gupta & England, 2005*; *Morandini et al., 2005*; *Ki et al., 2008*; *Bayha et al., 2010*; *Ramšak, Stopar & Malej, 2012*; *Dong, Liu & Liu, 2015*; *He et al., 2015*; *Chiaverano, Bayha & Graham, 2016*; *Scorrano et al., 2016*; *Gómez-Daglio & Dawson, 2017*; *Dong, 2018*.

**Type material:** Holotype: Medusa, MZUSP 8644. Paratypes: Medusa, MZUSP 8643; Polyps, MZUSP 8646; Tissue (Ephyrae), MZUSP 8645.

**Type locality:** Pedra do Baleeiro at Praia do Cabelo Gordo, São Sebastião, São Paulo, Brazil (Fig. 10).

**Etymology:** Named after the Centro de Biologia Marinha (CEBIMar) of the University of São Paulo, situated exactly where the type specimen was collected. This center is an international reference in marine biology studies, and many of the authors in this study have depended heavily on these facilities for their education and research.

**Distribution:** Our records include specimens from across the São Paulo state and from Aracaju, Sergipe. Therefore, the distribution likely spans the Brazilian coast from southeast to northeast (Table S4).

**Diagnosis:** There were five diagnostic positions for 16S, four for COI and 16 for ITS1 (Table S6).

**Remarks:** *Mayer (1910)* had identified specimens from the Brazilian coast as *A. aurita* and further records in the literature followed this classification (*Pantin & Dias, 1952*; *Vannucci, 1957*; *Goy, 1979*; *Mianzan & Cornelius, 1999*). However, we cannot confirm their identity to *A. cebimarensis* sp. nov. or others that occur or might occur in the country, and we therefore abstain from including as synonymous.

*Gambill & Jarms (2014)* had identified *Aurelia* polyps from the Brazilian coast (Ilha Grande, Rio de Janeiro), which had overall more tentacles than other populations, as *A.* sp. 2. However, sequences retrieved from these polyps, which came from the same locality and same culture as in their study, were recognized as a different species, *Aurelia insularia*

sp. nov. (see also remarks in this species' description herein). For a brief morphological description of the polyp, ephyra and medusa stages see Table S8.

***Aurelia malayensis*** Lawley, Gamero-Mora, Maronna, Chiaverano, Stampar, Hopcroft, Collins and Morandini **sp. nov.**

*Aurelia* sp. 4 *Dawson & Jacobs, 2001*. *Dawson, 2003*; *Dawson, Gupta & England, 2005*; *Ki et al., 2008*; *Ramšak, Stopar & Malej, 2012*; *Bayha & Graham, 2014*; *Dong, Liu & Liu, 2015*; *He et al., 2015*; *Chang et al., 2016*; *Chiaverano, Bayha & Graham, 2016*; *Scorrano et al., 2016*; *Dong et al., 2017*; *Abboud, Gómez-Daglio & Dawson, 2018*; *Dong, 2018*.

**Type material:** Holotype: Polyps, MZUSP 8650. Paratypes: Tissue (Polyps), MZUSP 8651.

**Type locality:** Palawan, Philippines (Fig. 10).

**Etymology:** Named after the Malay Archipelago, situated between mainland Indo-China and Australia, which includes the type locality and the suggested endemic distribution for this species (*Dawson, Gupta & England, 2005*).

**Distribution:** Across the Malay Archipelago to southern Japan, as well as in Hawaii (Table S4).

**Diagnosis:** There were 20 diagnostic positions for 16S, 17 for COI, 18 for ITS1 and four for 28S (Table S6).

**Remarks:** This species has been hypothesized as endemic to eastern Borneo and Palau, with the possibility of natural dispersal across the Malay Archipelago, south to Australia and north to Japan (*Dawson, Gupta & England, 2005*). Therefore, the occurrence in Hawaii would come from an anthropogenic introduction, likely after considerable WWII naval traffic (*Dawson, Gupta & England, 2005*).

Previous studies have shown that even though there are genetic differences distinguishing Palau specimens from those of other localities, they have similar rates of feeding, growth, respiration and swimming, if compared to *A. aurita* from the Black Sea (*Dawson & Martin, 2001*). Also, morphological variation between populations within a species sometimes exceeded variation between species within the Palau region (*Aurelia* sp. 3, *Aurelia* sp. 4, herein described as *A. malayensis* sp. nov., and *Aurelia* sp. 6, herein synonymized under *Aurelia clausa*; *Dawson, 2003*), which makes morphological diagnosis unreliable, as we also present in this study (further discussed).

*Mayer (1910)* mentions the distribution of *Aurelia colpota Brandt, 1835* across the Indo-Pacific. As the type specimen was described in South Africa and we cannot rely on morphology for further comparisons (further discussed), we refrain from resurrecting this name. Also, a COI sequence of a specimen from Palau, previously considered as *A.* sp. 4, was herein treated as *A.* cf. *malayensis*, as it appeared as sister taxa to the clade containing the remainder of the species' terminal taxa in both COI (Fig. S5) and concatenated phylogenies (Fig. 9; see Table S4 for details on this specimen's sequence). For a brief morphological description of the polyp, ephyra and medusa stages see Table S8.

***Aurelia clausa*** *Lesson, 1830*

*Aurelia* sp. 6 *Dawson & Jacobs, 2001*. *Dawson, 2003*; *Dawson, Gupta & England, 2005*; *Ki et al., 2008*; *Häussermann, Dawson & Försterra, 2009*; *Ramšak, Stopar & Malej, 2012*; *Dong, Liu & Liu, 2015*; *He et al., 2015*; *Chang et al., 2016*; *Chiaverano, Bayha & Graham, 2016*; *Scorrano et al., 2016*; *Abboud, Gómez-Daglio & Dawson, 2018*; *Dong, 2018*.

**Type material:** Based on our inquiries, no type material remains. Other material from the vicinity of the type locality (New Britain, Papua New Guinea) might remain in the private collection of *Dawson, Gupta & England (2005)*, as they deposited a sequence from this locality in GenBank (Table S4).

**Type locality:** New Ireland, Papua New Guinea (Fig. 10).

**Distribution:** Palau lakes, Papua and Papua New Guinea (Table S4).

**Diagnosis:** There were 20 diagnostic positions for COI and 20 for ITS1 (Table S6).

**Remarks:** Some sequences from New Zealand posted in GenBank were identified as *Aurelia* aff. *clausa* (Table S4). However, *A. clausa* was described from New Ireland, Papua New Guinea, and a specimen that belongs to the previously considered *A.* sp. 6 lineage was collected from New Britain, Papua New Guinea, in the vicinity of the type locality. Therefore, specimens in this lineage are here synonymized under *A. clausa*. For more information on previous studies regarding ecology and morphology of this species see remarks for *A. malayensis* sp. nov. For a brief morphological description of the medusa stage see Table S8.

***Aurelia dubia*** *Vanhöffen, 1888*

*Aurelia* ARAB lineage *Schroth et al., 2002*. *Dawson, 2003*; *Dawson, Gupta & England, 2005*.

**Type material:** Based on our inquiries, no type material remains. Other material might remain in the private collection of *Schroth et al. (2002)*, as they deposited the sequences for this species in GenBank (Table S4).

**Type locality:** Persian (Arabian) Gulf (Fig. 10).

**Distribution:** Arabian Peninsula, in the Red Sea and Persian Gulf (Table S4).

**Diagnosis:** There were eight diagnostic positions for 16S and 20 for ITS1 (Table S6).

**Remarks:** *Schroth et al. (2002)* defined the ARAB lineage with specimens from the Red Sea and from the Persian Gulf, the latter indicated as the type locality for *Aurelia dubia*. Nevertheless, they only deposited two sequences from this lineage in GenBank, one for 16S and one for ITS1, the former without any specification of the collection locality and the latter from a Persian Gulf specimen. In our single-marker phylogenies, the ITS1 sequence appears separate from all other *Aurelia* (Fig. S6), while for 16S, it forms a clade with a specimen from the Red Sea (Fig. S4). Considering that the ARAB lineage was defined also

based on samples from the Red Sea, it is possible that these specimens belong to the same species. We herein resurrect *A. dubia* encompassing the distribution of the ARAB lineage (suggested previously in *Dawson, 2003*), although we identify the specimen from the Red Sea as *A.* cf. *dubia*, until more markers are sequenced or further samples are collected that can ensure the identity of this specimen within *A. dubia*. For a brief morphological description of the medusa stage see Table S8.

***Aurelia solida*** *Browne, 1905*

*Aurelia* TET lineage *Schroth et al., 2002*.
*Aurelia* sp. 8 *Dawson, Gupta & England, 2005*. *Ramšak, Stopar & Malej, 2012*; *Ki et al., 2008*; *Manzari et al., 2015*; *Dong, Liu & Liu, 2015*; *He et al., 2015*; *Marques et al., 2014*; *Chiaverano, Bayha & Graham, 2016*.
*Aurelia* sp. *Tinta et al., 2010* (Bay of Piran).

**Type material:** Holotype: Medusa, NHM 1948.10.1.239.

**Type locality:** Republic of Maldives (Fig. 10).

**Distribution:** Across the Mediterranean Sea and the Red Sea (Table S4).

**Diagnosis:** Absence of an endodermal ocellus on the subumbrellar side of rhopalia (*Scorrano et al., 2016*; see remarks below). There were 10 diagnostic positions for 16S, 20 for COI, 20 for ITS1 and three for 28S (Table S6).

**Remarks:** The recent redescription of this species established holotypes and paratypes (*Scorrano et al., 2016*), but a holotype was already available and therefore is here designated. Also, the locality where specimens were collected for redescription is not concordant with the type locality (Republic of Maldives), so resurrection of the name was based on the direction of the rhopalium, which pointed to the exumbrellar side (90° angle) (*Scorrano et al., 2016*). However, we also observed this in specimens from very distinct localities, such as the southwestern USA and the Atlantic Ocean off Portugal (Figs. 12A–12D). Other observations herein have also indicated that morphology of rhopalia can vary even within species (Figs. 7B, 7D). Nevertheless, the presence or absence of an endodermal ocellus in specimens that also had an angled rhopalium could not be verified, as they can fade with preservation. This character may also vary, but until further specimens are analyzed, it is maintained as diagnostic alongside the genetic diagnosis.

No sequences have been obtained from specimens of the Maldives to confirm the distribution of this species in this locality. Nevertheless, it has been hypothesized that this species was introduced from the Indian Ocean into the Mediterranean through the Suez Canal (*Dawson, Gupta & England, 2005*; *Scorrano et al., 2016*). For a brief morphological description of the polyp, ephyra and medusa stages see Table S8.

***Aurelia labiata*** *Chamisso & Eysenhardt, 1821*

**Type material:** Neotype: Medusa, CASIZ 111024.

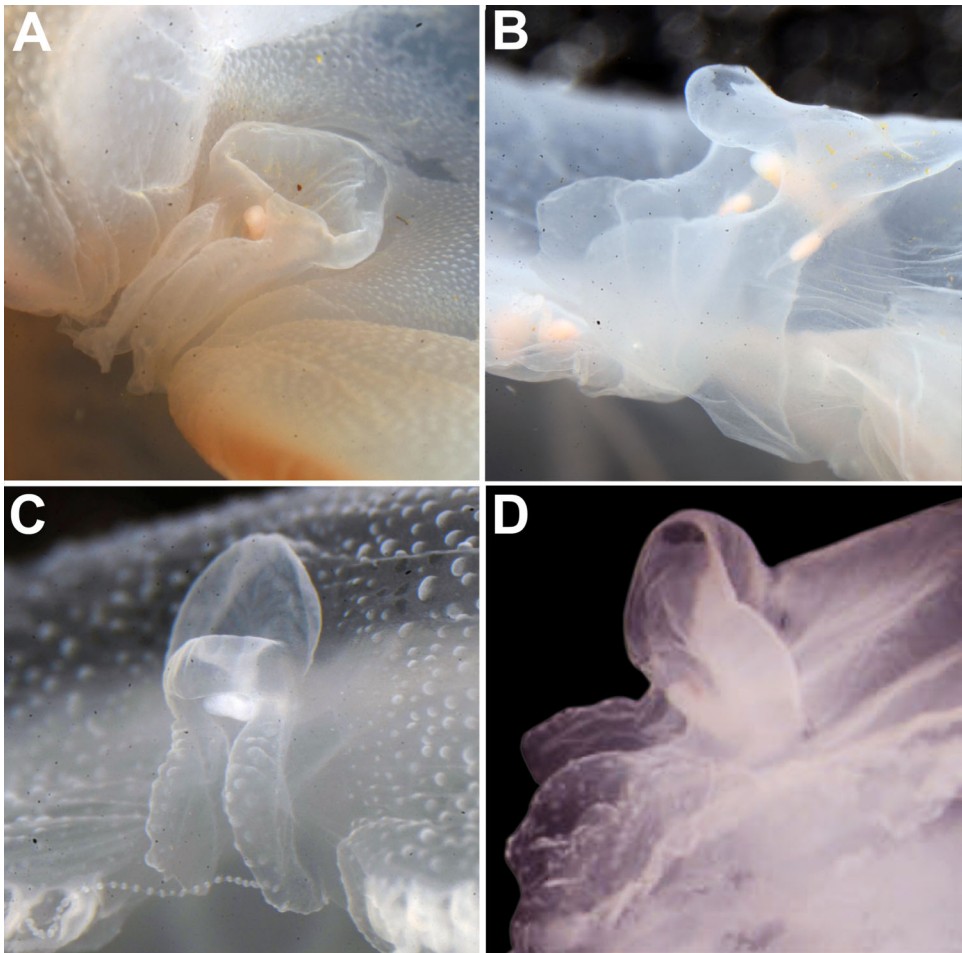

**Figure 12 Comparison of rhopalium morphology observed in some *Aurelia* medusae.** The 90° angled sense organ can be noticed in medusae from various localities, which includes the southwestern coast of the USA (USNM 92911-1, 92912-4) (A, B, respectively), the Atlantic Ocean off Portugal (USNM 58263-1) (C) and *Aurelia solida* (modified from *Scorrano et al., 2016*) (D). A, bell diameter (f1) = 13.45 cm; B, f1 = 10.6 cm; C, f1 = 5 cm; D, f1 = 14.4 cm.

**Type locality:** Monterey Bay, California, United States of America (Fig. 10).

**Distribution:** Northern coast of California, USA, north to Canada and into Alaska, USA (Table S4).

**Diagnosis:** There were 13 diagnostic positions for COI (Table S6).

**Remarks:** In the COI phylogeny we were able to observe two distinct clades within what was previously considered as *Aurelia labiata* (Fig. S5). In the 16S and ITS1 single-marker phylogenies, due to less sampling or even to different evolutionary rates across markers, it was not possible to observe reciprocally monophyletic clades (Figs. S4, S6). However, five specimens, two within *A. labiata* and three within *Aurelia columbia* sp. nov. (herein described) clades in the COI phylogeny had one other sequenced marker (Table S4). These specimens were used for the concatenated phylogenetic analysis, which

recovered the same reciprocal monophyly as for COI, as well as to tentatively identify 16S and ITS1 sequences that fell within the same monophyletic or paraphyletic group in these phylogenies (see *A.* cf. *labiata* and *A.* cf. *columbia* in Table S4).

In only one of the COI clades was there a specimen from California, USA (Fig. S5), where the type locality for *A. labiata* is situated, and is therefore included under this species' hypothesis. Also, additional preserved material in this species' redescription (*Gershwin, 2001*) is from Tomales Bay, California (CAS 111023), from where the Californian sequenced specimens included herein are. As the distribution of *A. labiata* overlaps with *A. columbia* sp. nov., we refrain from acknowledging any previous mentions as synonyms.

A distinct character included as diagnostic in both the original description and redescription of *A. labiata*, is the prominent manubrium (from the latin *labium*, meaning "lip"; for images and illustrations see *Gershwin, 2001*). This feature has been previously reported for other localities in the Pacific and Indian oceans, in specimens identified as *A. labiata* or even as *Aurelia maldivensis Bigelow, 1904* (*Mayer, 1910*). We also made these observations in some of the preserved specimens from the western coast of the USA, the Atlantic Ocean off Portugal and even from other localities, such as Japan and the western coast of Panama (Figs. 13A–13B; also see f2 in Tables S1–S2). Also, the number of marginal lobes (also called bell scalloping), considered previously as 16 for *A. labiata* and its variaties (*Mayer, 1910*), had already been disregarded as taxonomically significant in the species' redescription. This can be further emphasized in this study, as specimens from the Brazilian coast also seem to have more pronounced non-rhopaliar indentations (see f30 in Figs. 3–4), which defines the secondary scalloping.

None of the specimens sequenced from the western Pacific or Indian Oceans, which can present similar morphology to the previously considered *A. labiata*, clustered within any of the non-introduced northeastern Pacific species clades (which excludes *A. coerulea*). Until further studies can assess variability and plasticity of bell indentations and manubrium length, we refrain from using these characters in the diagnosis. For a brief morphological description of the polyp, ephyra and medusa stages see Table S8.

***Aurelia columbia*** Lawley, Gamero-Mora, Maronna, Chiaverano, Stampar, Hopcroft, Collins and Morandini **sp. nov.**

**Type material:** Holotype: Polyps, UF 12778.

**Type locality:** San Juan Island, Washington, USA (Fig. 10).

**Etymology:** Named after British Columbia, where most of the sequenced specimens have been collected.

**Distribution:** Northwestern coast of the USA, north to Canada (Table S4).

**Diagnosis:** There were 16 diagnostic positions for COI (Table S6).

**Remarks:** This species is sympatric with *A. labiata*, which spans from California to Alaska (USA) in the northeastern Pacific (see remarks for *A. labiata*). *Gershwin (2001)*, in the

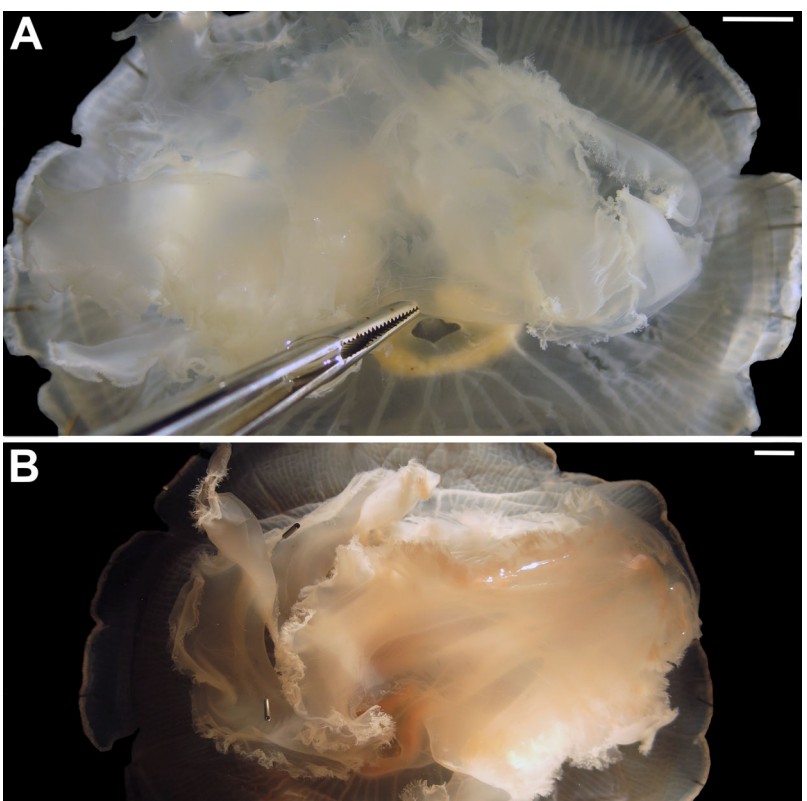

**Figure 13 Comparison of manubrium morphology observed in two *Aurelia* medusae.** The prominent manubrium, which sideways almost reached the margin of the umbrella, can be noticed in medusae from distinct localities, which includes the Atlantic Ocean off Portugal (USNM 58284) (A) and the south-western coast of the USA (USNM 92911-2) (B). Scales = 1 cm.

redescription of *A. labiata*, observed three morphotypes occurring in a latitudinal gradient. These morphotypes may not be species-specific, as we here describe another species that occurs across the range of *A. labiata*. Further studies integrating molecular phylogenetics and morphometrics may unravel morphological variation and plasticity within these species. For a brief morphological description of the polyp stage see Table S8.

***Aurelia relicta*** *Scorrano et al., 2016*

*Aurelia* sp. *Benovic et al., 2000*. *Malej et al., 2007*; *Turk et al., 2008*; *Tinta et al., 2010* (Big Lake).
*Aurelia* sp. 5 *Dawson & Jacobs, 2001*. *Dawson, Gupta & England, 2005*; *Ki et al., 2008*; *D'Ambra & Graham, 2009*; *Malej et al., 2009*; *Kogovšek et al., 2012*; *Korsun, Fahrni & Pawlowski, 2012*; *Ramšak, Stopar & Malej, 2012*; *D'Ambra et al., 2013*; *Manzari et al., 2015*; *Wang & Sun, 2014*; *Chiaverano, Graham & Costello, 2015*; *Dong, Liu & Liu, 2015*; *He et al., 2015*; *Marques et al., 2014*; *Chiaverano, Bayha & Graham, 2016*; *Miloslavić et al., 2016*.
*Aurelia* MS-MKL *Schroth et al., 2002*.

**Type material:** Holotype: Medusa, UNIPD CN57CH. Paratypes: Medusa, UNIS_SCY_028/29.

**Type locality:** Veliko Jezero, Mljet Island, Croatia (Fig. 10).

**Distribution:** Mljet Island lakes, Croatia (Table S4).

**Diagnosis:** There were eight diagnostic positions for 16S, 16 for COI, 20 for ITS1 and two for 28S (Table S6).

**Remarks:** In *Ramšak, Stopar & Malej (2012)*, one of the specimens collected in the Black Sea, in the Turkish coast, appeared as sister taxa to specimens from the West Atlantic, in their combined-marker phylogeny. In our single-marker phylogenies, we observed that the ITS1 sequence from that specimen fell within a clade alongside the other sequences from the same locality, within the *A. aurita* clade (Fig. S6), while the COI sequence from that same specimen fell within the *A. relicta* clade (Fig. S5). This can potentially be due to contamination in sequencing the COI, as in the aforementioned study, *A. relicta* specimens from the Mljet lakes were also being sequenced. As other sequences from both *A. aurita* and *A. relicta* were available, this specimen from *Ramšak, Stopar & Malej (2012)* was disregarded from the concatenated phylogenetic analysis in our study.

*Scorrano et al. (2016)* presented a table with diagnostic characters for some of the Mediterranean species of *Aurelia*, from the polyp, ephyra and medusa stages. Nevertheless, based on the unreliability of medusa morphometric features for species recognition shown herein (further discussed); and the potential confusion that can arise from polyp and ephyra morphology (*Gambill & Jarms, 2014*), especially considering the possibility of morphological plasticity in these life cycle stages (*Chiaverano & Graham, 2017*), we refrain from including them here. Furthermore, there seem to be no unambiguous categorical features, if compared to *A. coerulea* and *A. solida* (see Table 2 in *Scorrano et al., 2016*, and remarks of these species in this study). For a brief morphological description of the polyp, ephyra and medusa stages see Table S8.

***Aurelia persea*** (*Forskål, 1775*)

*Aurelia* sp. *Mizrahi, 2014*.

**Type material:** Based on our inquiries, no type material remains. Other material from the type locality region might remain in the private collection of *Mizrahi (2014)*, as he deposited sequences of a specimen from this locality in GenBank (Table S4).

**Type locality:** Mediterranean Sea (Fig. 10).

**Distribution:** Sequences of specimens herein included derive only from Haifa Bay, Israel (Table S4).

**Diagnosis:** There were 13 diagnostic positions for 16S, 20 for COI and three for 28S (Table S6).

**Remarks:** The original description of this species is brief and simple, which therefore later rendered it as synonymous to *A. aurita* (*Agassiz, 1862*). Even if the description was more informative, there is only one image of the sequenced specimen (see Fig. 17 in *Mizrahi, 2014*), from which hardly any information can be retrieved. Furthermore, as we portray in this study the unreliability of medusa morphology for species identification (further discussed), we resurrect *Aurelia persea* because it is the oldest available name that encompasses the locality of the sequenced specimen treated herein. For a brief morphological description of the medusa stage see Table S8.

**Aurelia aurita** (*Linnaeus, 1758*)

*Aurelia* BOR lineage *Schroth et al., 2002*.
*Aurelia borealis Schroth et al., 2002*.

**Type material:** Neotype: Tissue (Medusa), MZUSP 8657.

**Type locality:** Helgoland, Germany (Fig. 10).

**Distribution:** North, Black, Baltic and Caspian Seas, Northeast Atlantic, Greenland, northeastern USA and Canada, Northwest Pacific and South America (Table S4).

**Diagnosis:** There were five diagnostic positions for 16S, 20 for COI, 20 for ITS1 and 12 for 28S (Table S6).

**Remarks:** In *Ramšak, Stopar & Malej (2012)*, one of the specimens collected in the Mljet lakes, in Croatia, appeared as sister taxa to a specimen from the Southwest Atlantic in their combined-marker phylogeny. In our single-marker phylogenies, we observed that the ITS1 sequence from that specimen fell within a clade alongside the other sequences from the same locality, within the *Aurelia relicta* clade (Fig. S6), while the COI sequence from that same specimen fell within the *A. aurita* clade (Fig. S5). This can be due to contamination in sequencing the COI, as in the aforementioned study, *A. aurita* specimens were also being sequenced. As other sequences from *A. relicta* were available, this specimen from *Ramšak, Stopar & Malej (2012)* was disregarded for the concatenated phylogenetic analysis herein.

Previously, many species of *Aurelia* were synonymized under *A. aurita* (originally described from the Baltic Sea), as no morphological distinction could be made, and this species was considered globally distributed (*Kramp, 1965*, *1968*; *Russell, 1970*; *Larson, 1990*; *Arai, 1997*). More recently, it has been recognized that, alongside *A. coerulea*, this species has one the widest distributions in the genus, but possibly due to multiple introductions from its endemic range in the Northeast Atlantic (potentially naturally dispersed to northeastern USA, although not so likely; see *Dawson, Gupta & England, 2005*). Only one specimen of *A. aurita* is from the Northwest Pacific, reported from *Armani et al. (2013)*, from a northwestern Pacific sample that is also present in *Schroth et al. (2002)*. This could represent a new point of introduction of this species, and should be confirmed in the future with further collections in the area. Also, we recorded this species for the first time in Ushuaia, Argentina (Table S4), which could represent a new

point of introduction but also ongoing spread from a single introduction that has been recorded in other localities along that region of South America (*Häussermann, Dawson & Försterra, 2009*). Other synonyms for this species have been presented, but we refrain from maintaining them, as they could belong to other species and until now we could not confirm it. For a brief morphological description of the polyp, ephyra and medusa stages see Table S8.

### *Aurelia coerulea* von Lendenfeld, 1884

*Aurelia japonica* Kishinouye, 1891.
*Aurelia* sp. 1 *Dawson & Jacobs, 2001. Dawson, 2003*; *Dawson, Gupta & England, 2005*; *Ki et al., 2008*; *Häussermann, Dawson & Försterra, 2009*; *Ramšak, Stopar & Malej, 2012*; *Wang & Sun, 2014*; *Dong, Liu & Liu, 2015*; *He et al., 2015*; *Marques et al., 2014*; *Chiaverano, Bayha & Graham, 2016*; *Dong et al., 2017*.
*Aurelia* UBI lineage *Schroth et al., 2002*.
*Aurelia* sp. *Manzari et al., 2015*.

**Type material:** Holotype: NHM 1886.7.8.6.

**Type locality:** Port Jackson, Sydney, Australia (Fig. 10).

**Distribution:** Northwestern Pacific, Australia, west coast of the USA, Mediterranean and Atlantic coast of Europe (Table S4).

**Diagnosis:** There were 15 diagnostic positions for 16S, 20 for COI, 18 for ITS1 and one for 28S (Table S6).

**Remarks:** The recent redescription of this species established holotypes and paratypes (*Scorrano et al., 2016*), but a holotype was already available and therefore is here designated. Interestingly, the type locality is not concordant with the inferred biogeographic origin in the coastal waters of the Western Pacific (*Dawson, Gupta & England, 2005*). This species has one of the broadest distributions in the genus, with multiple introductions across the globe (*Dawson, Gupta & England, 2005*). Anecdotal observations of polyps in cultivation in different temperatures (15–24 °C), suggest that they strobilate more frequently than other *Aurelia* species, even though under the exact same conditions. This could enhance its potential for spread, a matter for future studies to test.

A potential distinct feature in this species is the dark-orange or brownish color of the recently released ephyrae, which is appointed as diagnostic (*Scorrano et al., 2016*) and that we have also observed in our lab cultures. However, until a further assessment of ephyrae coloration in more *Aurelia* species is undertaken, and due to past reports of the unreliability of coloration for species identification in this genus (*Kramp, 1968*) and in other Medusozoa (*Lampert et al., 2011*; *Holst & Laakmann, 2014*), we abstain from including this as diagnostic. Further characters also indicated as diagnostic for polyps and ephyrae can derive from morphological variability, which has been noticed in this species (*Scorrano et al., 2016*) and also in other species of the genus (*Gambill & Jarms, 2014*; *Chiaverano & Graham, 2017*). For more information on morphological variability in

medusae of this species see section 'Morphological variability and diagnosis in *Aurelia coerulea*'. For a brief morphological description of the polyp, ephyra and medusa stages see Table S8.

***Aurelia limbata*** *Brandt, 1835*

**Type material:** Neotype: Polyps, MZUSP 8660. Paraneotype: Tissue (Polyps), MZUSP 8661.

**Type locality:** Okirai Bay, Ofunato, Japan (Fig. 10).

**Distribution:** Northwestern Pacific (Table S4).

**Diagnosis:** There were three diagnostic positions for 16S, 20 for COI, 19 for ITS1 and six for 28S (Table S6).

**Remarks:** *Brandt (1835)* described this species of *Aurelia* from the northwestern Pacific (Avacha Bay, Kamchatka, Russia) as very distinct due to the dark-brownish color of its bell margin and the brown or yellowish coloration of radial canals, which were highly ramified. This is clearly represented in the illustration in his next publication (*Brandt, 1838*). This morphological pattern is also associated to records in the northeastern Pacific, including the cover photograph of the January 1974 issue of Audubon magazine, featuring a specimen from the Aleutian Islands (*Larson, 1990*; *Gershwin, 2001*). However, more recent accounts, including the sequences herein, are only from the northwestern Pacific (*Miyake et al., 2002*; *Chang et al., 2016*).

There are other *Aurelia* species that occur in the northeastern Pacific, in Alaska, USA, such as *Aurelia hyalina* and *A. labiata*, the former even previously identified as *A. limbata* (see remarks for *A. hyalina* in this study). *Gershwin (2001)* even suggested that *A. limbata* could be a color morph, part of the *A. labiata* species complex. Whether the distribution of *A. limbata* actually extends across the North Pacific or the distinct coloration is not intraspecific, is still unclear. Considering this controversy and previous accounts on the unreliability of coloration for species recognition in this genus (*Kramp, 1968*) and in other Medusozoa (*Lampert et al., 2011*; *Holst & Laakmann, 2014*), we refrain from including this as diagnostic.

Regarding the highly ramified radial canals in the original description (*Brandt, 1835*), we observed the highest number of branching points in specimens from Japan and Arctic Alaska, USA (black squares in Figs. 3–4; Fig. 6C). This is concordant with the distribution of sequenced specimens of *A. limbata* (Japan) and *A. hyalina* (Arctic). Therefore, as discussed previously for coloration, the ramification pattern of radial canals might not be intraspecific, and once more we refrain from including this in the diagnosis. This follows the conclusions of this study, that shows the unreliability of morphology for species recognition due to morphological variability (further discussed), and we present a molecular genetic diagnosis to support this species hypothesis.

In this potential confusion regarding distribution and morphology of *A. limbata* and *A. hyalina*, we abstain from reporting previous accounts as synonyms. Even with more

recent studies that use molecular data, such as *Schroth et al. (2002)*, there might be some issues. The 16S sequence these authors posted in GenBank from the LIM lineage, which they consider *A. limbata*, belongs to the Mljet lakes, Croatia, and therefore in our 16S phylogeny is part of the *A. relicta* clade (Fig. S4). Another issue is the LIM lineage ITS1 sequence posted in GenBank, which if submitted to NCBI's BLAST returns *Cyanea capillata* (*Linnaeus, 1758*) as the most similar taxon, the chosen outgroup in that study. These issues are not uncommon, and can derive from contamination or even sample mislabeling. Still, within the LIM lineage there are specimens from Iceland, but they were not deposited in GenBank, and therefore we cannot confirm their identity to the species clades treated herein, likely either *A. limbata* or *A. hyalina*. For a brief morphological description of the polyp, ephyra and medusa stages see Table S8.

**Aurelia hyalina** *Brandt, 1835*

*Aurelia limbata* *Dawson & Jacobs, 2001*.
*Aurelia* sp. 10 *Dawson, Gupta & England, 2005*. *Ki et al., 2008*; *Häussermann, Dawson & Försterra, 2009*; *Ramšak, Stopar & Malej, 2012*; *Dong, Liu & Liu, 2015*; *He et al., 2015*; *Scorrano et al., 2016*; *Dong, 2018*.

**Type material:** Neotype: Tissue (Medusa), USNM 1643575. Paraneotypes: Tissue (Medusa), USNM 1643579-1643580.

**Type locality:** Chukchi Sea, Russia (Fig. 10).

**Distribution:** Northeastern Russia, southwestern Alaska (USA), northeastern Canada and Greenland (Table S4).

**Diagnosis:** There were 12 diagnostic positions for COI and 19 for ITS1 (Table S6).

**Remarks:** In our 16S single-marker phylogeny this species appeared as a paraphyletic group (Fig. S4), although in the COI, ITS1 (Figs. S5–S6) and, more importantly, the concatenated phylogenies (Fig. 9), *A. hyalina* appears as monophyletic and sister to *A. limbata*.

We resurrect this name based on the neighboring distribution of the previous type locality (Aleutian Islands, Alaska, USA) with one of the sequenced specimens herein studied. This specimen was once considered to belong to *A. limbata* (*Dawson & Jacobs, 2001*), but later delimited as another species once other sequences from Japan and South Korea were added, which derived from specimens that fit within the original description of *A. limbata* (*Dawson, Gupta & England, 2005*; *Chang et al., 2016*; also see remarks for *A. limbata* in this study). For a brief morphological description of the medusa stage see Table S8.

**Undescribed species and other currently valid names**
Most of the species hypotheses in this study that remain undescribed had been previously noted and in some cases delimited, such as *A.* sp. 3 (*Dawson & Jacobs, 2001*), *Aurelia* sp. 7 (*Dawson, Gupta & England, 2005*), *A.* sp. 12, *A.* sp. 13, *A.* sp. 14 (*Gómez-Daglio & Dawson,*

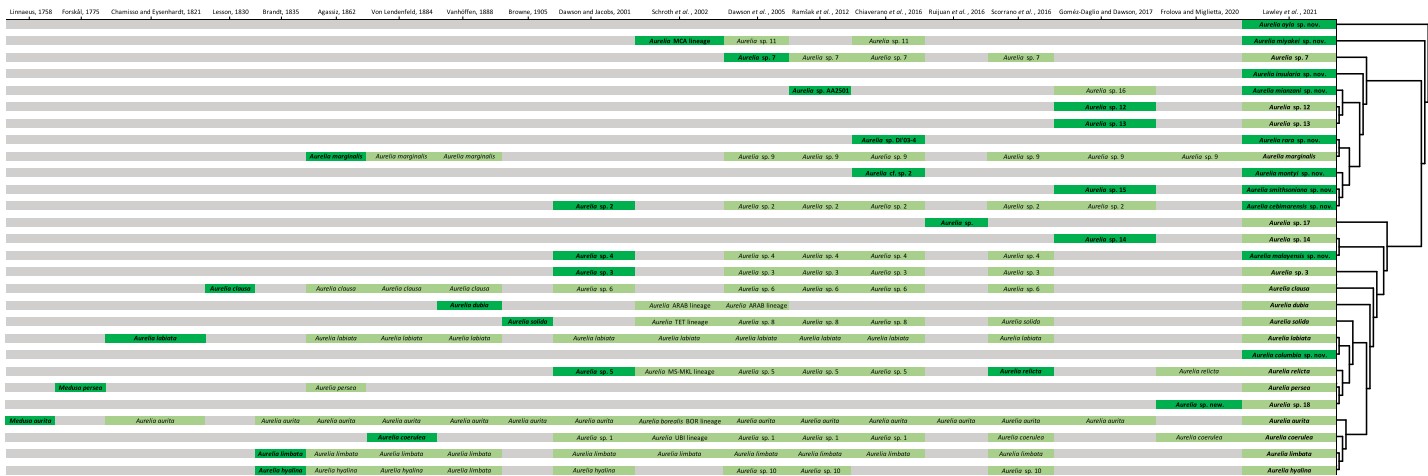

**Figure 14 Taxonomic history of *Aurelia* species treated herein.** All studies that delimited and/or described a species for the first time were included, and each of these instances is highlighted in dark green. Mentions of other species in these studies are highlighted in light green, including the name that was used. Species are ordered as terminals of the concatenated phylogeny (Fig. 9).

*2017*) and *Aurelia* sp. 18 (as *Aurelia* sp. new., *Frolova & Miglietta, 2020*). In the case of *Aurelia* sp. 17 from the western coast of Thailand, *Ruijuan et al. (2016)* identified it as *Aurelia* sp., as in their 18S phylogeny these specimens were in the same clade as an *A. coerulea* specimen (as *Aurelia* sp. EU276014), but in their 16S phylogeny, similar to the 16S (Fig. S4) and concatenated phylogenies presented herein (Fig. 9), it appears in a distinct clade.

In the most recent accounts of *Aurelia* species, three other names are currently valid that were not detected in this study (*A. colpota*, *A. maldivensis* and *Aurelia vitiana Agassiz & Mayer, 1899*; *Jarms & Morandini, 2019*; *Collins, Jarms & Morandini, 2020*). In these recent accounts, *A. dubia* was synonymized under *A. colpota*, but as sequenced specimens studied herein closely matched the type locality of *A. dubia*, we prioritise the resurrection of this epithet (see remarks for *A. dubia*). For *A. maldivensis* and *A. vitiana*, described from the Maldives and Fiji respectively, none of the distributions of the undescribed species hypotheses overlapped or were near their type localities, and we therefore avoided any synonymization. We recommend that each of these three names are treated as *species inquirenda*, pending further investigation. A summary of the taxonomic history of *Aurelia* species treated herein is provided in Fig. 14, showing studies that either described or delimited one of these species for the first time and what other species were mentioned, including what names were used.

## DISCUSSION

### Variability and the use of morphology to diagnose

Most descriptions of *Aurelia* species were based on the medusa stage, which is the most conspicuous and easily collected of the life cycle stages (*Mayer, 1910* reviews morphology of *Aurelia* species and their varieties that had been described). Overlaps in morphological differences across large spatial scales created much confusion for species

identification, until the incorporation of molecular genetic data propelled a re-evaluation of morphological characters in all life cycle stages (*Dawson & Jacobs, 2001*; *Schroth et al., 2002*; *Dawson, 2003*; *Dawson, Gupta & England, 2005*; *Gambill & Jarms, 2014*). Recent descriptions of species hypotheses based on genetic data acknowledge some morphological features of medusae as diagnostic (*Scorrano et al., 2016*).

In our evaluation of *Aurelia* specimens from across the globe, we found no geographic organization associated with morphological variation. On the contrary, morphological variation among specimens within regions, and even within collection lots, usually overlapped with that of specimens from very distinct localities (Figs. 3–4). If, however, neighboring regions had structured morphological dissimilarities, this could mean that morphotypes might be distinguished in smaller spatial scales, and if related to species hypotheses, these could be useful to distinguish neighbor or even sympatric species (*e.g.*, *Aurelia* in the Gulf of Mexico, as in *Chiaverano, Bayha & Graham, 2016*). We did not observe this pattern, but the opposite, that specimens distributed closer to each other tended to be more similar, although this was presented as either a weak or non-significant relationship. The morphological variation discussed above also encompassed previously considered categorical features, which could not be unambiguously determined, likely due to the higher sampling effort of this study, and were either removed from analyses or adapted to continuous or meristic features (such as f34-36 reflecting continuous variation from previously used f8; Table S3).

The comparison of cultured *Aurelia coerulea* medusae with the species diagnosis, provided by specimens studied in the Mediterranean (*Scorrano et al., 2016*), further illustrates the potential for morphological variability, in both continuous and categorical features (illustrated in Fig. 7). Interestingly, the only specimens analyzed that were more morphologically similar to each other were from the cultures at the Discovery Place Aquarium (DP-Aq in Figs. 3–4; identified as *A. coerulea* based on genetic data, see Table S4). These were raised under roughly the same controlled conditions, such as temperature, water circulation, light intensity, and fed the same amount at the same time. Controlled conditions that reflect a certain morphological pattern corroborates the hypothesis for environmentally determined morphological plasticity, which has already been demonstrated for medusae of an *Aurelia* species in the field (*Chiaverano, Bayha & Graham, 2016*). All of the evidence mentioned above favors the argument that medusa morphology is likely uninformative for species diagnosis in this genus.

To further complicate matters, there are hypothesized multiple introductions of *Aurelia* species across the globe (*Dawson, Gupta & England, 2005*; see examples in the remarks for *A. coerulea*, *Aurelia aurita* and *Aurelia solida*), and likely more still undetected. Even if species within neighboring regions could be distinguished by morphology, introduced species could confuse these distinctions. This could also have confused our morphological analysis, as it is based on the geographic distribution of morphological dissimilarities that, in most cases, did not have direct equivalence to the genetic dataset, in which species hypotheses were based. Still, even considering potential confusions from that scenario, by relating the determined geographic regions with sampling sites of species hypotheses from genetic markers, no structure appears from

morphological data (*e.g.*, Japan and USA-SW, which could both belong to *A. coerulea*, see Figs. 3–4).

However uninformative medusa morphology may be for species distinction, it is interesting to ponder the characters that account for most of the morphological variation across specimens, such as the branching pattern of radial canals and bell indentations, the latter which determines the number of lobes (scallops) on the umbrella margin. These characters were some of those previously used to recognize a few species: *Aurelia labiata* and *Aurelia limbata* were distinguished by the possession of 16 marginal scallops, while *A. aurita* only had eight (*Mayer, 1910*; *Gershwin, 2001*); *A. limbata* was also reported to have highly branched radial canals in comparison to other species (*Mayer, 1910*; *Gershwin, 2001*). As more specimens were collected through time, these distinctions started to fade, and are further discussed for each species, when applicable, in the remarks of their systematic account in this study. Only one character from the medusa stage was maintained as potentially diagnostic, the absence of the endodermal ocellus in the rhopalia of *A. solida* (*Scorrano et al., 2016*). This character is usually faded in preserved material, and we could not observe it in the museum specimens analyzed.

Other candidates as diagnostic morphological characters derive from other stages of the life cycle, such as polyps and ephyrae (*Gambill & Jarms, 2014*; *Scorrano et al., 2016*), which were not the focus of the morphological assessment herein presented. Nevertheless, previous studies have compared them in *Aurelia*, and have shown the overlap in morphology of these stages in different hypothetical species (*Gambill & Jarms, 2014*), as well as morphological plasticity in different sets of controlled conditions (*Chiaverano & Graham, 2017*), in line with the overall patterns discussed here for the medusa stage. Only one morphological character was here maintained as potentially diagnostic, the higher number of tentacles in polyps of *Aurelia insularia* sp. nov. (as *Aurelia* sp. 2 from *Gambill & Jarms, 2014*), until further studies can re-address this more thoroughly across the recently recognized diversity. Further discussions on the morphology of polyps and ephyrae, when applicable, are present in the remarks of each species' description.

## Species delimitation, cryptic diversity and the transition to species description

Acknowledging that morphology may not be informative for taxonomy, at least for some groups of metazoans, can be daunting. Morphology has been the basis of taxonomy for centuries, although the increase in accessibility to genetic data has raised doubts (*Dayrat, 2005*). Many studies that embrace this new source of information have revealed a previously undetected diversity, mostly named as 'cryptic' (for a review see *Bickford et al., 2007*). In result, it has been suggested that molecular data could be the only solution to assess the planet's biodiversity in the midst of the extinction of both species and taxonomists (*Hebert et al., 2003*). Even though there is little consensus in that view (made clear by the reviews and comments in *Goldstein & DeSalle, 2011*, *Collins & Cruickshank, 2013* and *DeSalle & Goldstein, 2019*), few studies have accepted the challenge of reconciling species delimitation and description for medusozoans, thus failing to

provide both the scientific community and society of this taxonomic service (*Jörger & Schrödl, 2013*).

Prior to descriptions, we assessed the use of molecular genetic markers herein studied as barcodes, in the sense of a potential tool for rapid identification. COI remains as the best candidate (as previously suggested for most metazoans, as well as medusozoans; *Hebert et al., 2003*; *Ortman et al., 2010*), as there is a greater gap between most intra- and interspecific distances (Fig. 11). However, some of the hypothesized species have between 6.2% and 9.8% of differences between them, and in the case of the sister species *Aurelia smithsoniana* sp. nov. and *Aurelia cebimarensis* sp. nov. it was as low as 2% (Fig. 11; Table S7). As evolutionary rates may vary across congeners, it is hard to set a threshold for species identification, and this gap could be partly an artifact of unknown diversity due to undersampling (*Meyer & Paulay, 2005*; *Wiemers & Fiedler, 2007*; see *Gómez-Daglio & Dawson, 2017* for other examples in medusozoans). Also, species hypotheses may change with future studies, and this gap could become more or less pronounced depending on what species hypotheses are accepted and considered. This may be a useful tool for first assessments and the discovery of potentially cryptic species, but it might not be reliable for species identification. Even less so should it be used for species delimitation, as neo-phenetic arbitrary constructs should not replace testable species hypotheses (*Prendini, 2005*; *Valdecasas, Williams & Wheeler, 2008*; see more in the 'Materials and Methods' section). For quite some time now the scientific community has accepted that similarity does not necessarily reflect kinship (*i.e.*, evolutionary relationship), one of the basic principles of phylogenetic systematics (*Hennig, 1966*), which remains a key component for molecular species delimitation and taxonomy (*Gómez-Daglio & Dawson, 2017*).

With the results from past studies and those provided herein, we demonstrate that morphology is likely uninformative for distinguishing at least most of the species of the *Aurelia* genus. Even though some characters might yet be revealed as useful, and as we are only beginning to understand morphological variability and diversity within the genus, providing formal descriptions with a character-based diagnosis seems paramount to develop a taxonomic foundation for future studies. Character-based diagnosis, molecular or not, provides a falsifiable and comparable basis in which to build species hypotheses and descriptions (*Grant et al., 2006*; *Bauer et al., 2011*) and is required by the *ICZN (International Commission on Zoological Nomenclature) (1999)*; Article 13.1.1.). Also required to accompany newly described species are name-bearing types (*ICZN (International Commission on Zoological Nomenclature), 1999*; Article 72.3). Ideally, the type material that accompanies newly described species should be a specimen, from which a subsample is taken and DNA is extracted. For some samples in this study that was not possible, so to comply with the ICZN, the type material is provided as tissues or DNA extractions, and further specimens from the same culture or collection (when no sympatry had been recorded), when available, were provided within the type series (for other examples of species descriptions with molecular genetic diagnosis and tissues or DNA extractions as type material, see *Jörger & Schrödl, 2013*; *Eitel et al., 2018*).

Diagnostic molecular genetic characters have been identified as, for example, character attributes from sequence alignments, with sequences manually identified in groups of previously determined species hypotheses (*Sarkar, Planet & DeSalle, 2008*; as in *Jörger & Schrödl, 2013*), or as synapomorphies for the species clades observed in a phylogenetic tree (*Machado, 2015*; *Eitel et al., 2018*). We reported diagnostic characters as synapomorphies (*sensu Grant & Kluge, 2004*), as these rely directly on a phylogenetic inference and are portrayed in categories defined based on all possible optimization schemes for character-states (output from the program YBYRÁ; *Machado, 2015*). As a result, synapomorphies can be classified as ambiguously or unambiguously optimized, the latter which is further categorized into unique and non-homoplastic, unique and homoplastic or non-unique and homoplastic (*Machado, 2015*). The desired scenario regarding these categories would be to have unique and non-homoplastic synapomorphies (in black, Fig. 9 and Table S6) for each species hypothesis. With only five possible character-states (gaps as fifth state), it is not surprising that many species did not present these synapomorphies, mostly for COI, which is likely related to the much greater number of sequences in this dataset but could also be associated with varying evolutionary rates across markers. In that sense, the combination of synapomorphies as diagnostic, regardless of the category, could be more reliable.

There seems to be great potential in synapomorphies not only to construct species hypotheses and provide descriptions, but also for species identification. A synapomorphy-based identification can be much more reliable than conventional barcoding or NCBI's BLAST, as it is not based on similarity but on specified characters that directly reflect species hypotheses. This has been somewhat attempted with CAOS's P-Elf program (*Sarkar, Planet & DeSalle, 2008*), but to our knowledge, none of the authors that report diagnostic characters from this program, such as *Jörger & Schrödl (2013)* and *Maggioni et al. (2017)*, provided the output of the program's P-Gnome module, which would be used for classifying new sequences. These authors have otherwise suggested that diagnostic synapomorphy positions from the alignment, retrieved from CAOS, should be mapped to a reference sequence and both positions reported in the description. Yet, if other researchers seek to manually map their newly acquired sequences with any of the suggested above, for species identification, insertions and deletions could highly confuse the process, especially in genetic markers that commonly present them, such as those from ribosomal RNA regions (*e.g.*, 16S, ITS1 and 28S). Furthermore, the algorithm used by P-Elf to classify new sequences is not clearly stated (*Sarkar, Planet & DeSalle, 2008*). A prospect for future studies would be to better evaluate and understand the possible issues involved in synapomorphy-based identifications and how to convert them in a computational pipeline that can be easily and widely used, such as the BLAST tool.

## CONCLUSIONS

Our conclusion with this study is not that morphology should be left aside. On the contrary, we are just beginning to unravel how morphological variation can be

environmentally induced (*Chiaverano & Graham, 2017*), as well as the evolutionary processes involved in morphological change and speciation (see *Struck et al., 2017*). For example, the morphological overlap we observed across species could be related to recent divergences, parallelism, convergence or even stasis, and most of these have already been demonstrated to occur in other medusozoans (*Swift, Gómez-Daglio & Dawson, 2016*). A starting point for such studies in the jellyfish genus *Aurelia* could be investigating the characters that accounted for most of the morphological variation detected herein, such as bell indentations and ramification of radial canals, on more fine spatial scales considering environmental variation. This next step is fundamental to understand mechanisms that generated biodiversity and how these could be impacted by future changes.

## ACKNOWLEDGEMENTS

We thank institutions and associated staff that kindly provided us with preserved specimens for this study, such as Dr. Adam Baldinger (MCZ), Dr. John Slapcinsky (FLMNH), Dr. Eric Lazo-Wasem and Dr. Lourdes Rojas (YPM), Dr. Laura Pavesi (ZMUC), Prof. Elizabeth Neves (UFBA/MZUFBA), Dr. Aline Benetti (MZUSP), and Jorge Thé de Araújo (UFC). Also, a special appreciation to the Smithsonian Institution and the Invertebrate Zoology Department staff, William Geoff Keel, William Moser, Courtney Wickel, Chad Walter, Freya Goetz, Dr. Abigail Reft and Linda Cole, for providing not only specimens but support for analyzing most of them. Also, the aquariums and their staff that contributed with live material, Matt Wade (NA) and Matt Lowder (DP), as well as Dr. Jason Macrander and the Reitzel lab at University of North Carolina Charlotte (UNCC) for kindly assisting with the visit to DP. Molecular analyses were made possible by the Laboratory of Molecular Evolution at the University of São Paulo and their staff Manuel Antunes Jr., Beatriz V. Freire and Dr. Sabrina Baroni. Prof. Sérgio Tadeu helped with insights in the multivariate statistical analyses. We also thank Prof. Michael N. Dawson for comments on the first version of this manuscript, as well as the editor and three reviewers for suggestions on improvements prior to publication.

### Funding

This work was supported by Fundação de Amparo à Pesquisa do Estado de São Paulo (2015/21007-9, 2016/04560-9, 2016/12163-0, 2017/07317-0, 2019/03552-0), Conselho Nacional de Desenvolvimento Científico e Tecnológico (133900/2016-9, 309440/2019-0) and Coordenação de Aperfeiçoamento de Pessoal de Nível Superior (238.273.628-30). The funders had no role in study design, data collection and analysis, decision to publish, or preparation of the manuscript.

## Grant Disclosures

The following grant information was disclosed by the authors:
Fundação de Amparo à Pesquisa do Estado de São Paulo: 2015/21007-9, 2016/04560-9, 2016/12163-0, 2017/07317-0, 2019/03552-0.
Conselho Nacional de Desenvolvimento Científico e Tecnológico: 133900/2016-9, 309440/2019-0.
Coordenação de Aperfeiçoamento de Pessoal de Nível Superior: 238.273.628-30.

## Competing Interests

The authors declare that they have no competing interests.

## Author Contributions

- Jonathan W. Lawley conceived and designed the experiments, performed the experiments, analyzed the data, prepared figures and/or tables, authored or reviewed drafts of the paper, and approved the final draft.
- Edgar Gamero-Mora performed the experiments, prepared figures and/or tables, authored or reviewed drafts of the paper, and approved the final draft.
- Maximiliano M. Maronna performed the experiments, prepared figures and/or tables, authored or reviewed drafts of the paper, and approved the final draft.
- Luciano M. Chiaverano performed the experiments, authored or reviewed drafts of the paper, and approved the final draft.
- Sérgio N. Stampar conceived and designed the experiments, performed the experiments, authored or reviewed drafts of the paper, and approved the final draft.
- Russell R. Hopcroft performed the experiments, authored or reviewed drafts of the paper, and approved the final draft.
- Allen G. Collins conceived and designed the experiments, performed the experiments, authored or reviewed drafts of the paper, and approved the final draft.
- André C. Morandini conceived and designed the experiments, authored or reviewed drafts of the paper, and approved the final draft.

## DNA Deposition

The following information was supplied regarding the deposition of DNA sequences:
The sequences are available at GenBank: MZ061727–MZ061824, MZ061859–MZ061865, MZ061825–MZ061858, MZ052131–MZ052208.
The trees and alignments are available at Figshare: Lawley, Jonathan; Gamero-Mora, Edgar; Maronna, Maximiliano; M. Chiaverano, Luciano; N. Stampar, Sergio; Hopcroft, Russell R.; et al. (2021): Genetic data for Aurelia systematics. figshare. Dataset. DOI 10.6084/m9.figshare.14502474.v1.

## Data Availability

Raw data and codes are only available in the mentioned GitHub page. They were provided with the manuscript submission as "review-only".

## New Species Registration

The following information was supplied regarding the registration of a newly described species:

Publication LSID: http://zoobank.org/9CCDC703-92EB-4EDD-AB8F-F353941FEA1B.
*Aurelia ayla* sp. nov.: urn:lsid:zoobank.org:act:F055C0AE-0FF0-46BB-965D-10510EF6B21D, *Aurelia insularia* sp. nov.: urn:lsid:zoobank.org:act:4EB4EA19-5C65-4F5A-8773-AB3ECADEB7B5, *Aurelia mianzani* sp. nov.: urn:lsid:zoobank.org:act:1B75D6DC-59B2-4038-88A1-94E814FBD46E, *Aurelia miyakei* sp. nov.: urn:lsid:zoobank.org:act:7F1A440F-52E4-4720-B663-E8675C131B66, *Aurelia rara* sp. nov.: urn:lsid:zoobank.org:act:EC675BB6-36FA-4A7A-936C-882D92587CE4, *Aurelia montyi* sp. nov.: urn:lsid:zoobank.org:act:AB98C8F2-A1EC-4E8F-BCE7-F5FA8012193F, *Aurelia smithsoniana* sp. nov.: urn:lsid:zoobank.org:act:811D0D34-179B-49C7-9DE2-C43A8F5C7A72, *Aurelia cebimarensis* sp. nov.: urn:lsid:zoobank.org:act:BE97F71E-5E61-4E9B-AEC5-DAF2FA03C9C0, *Aurelia malayensis* sp. nov.: urn:lsid:zoobank.org:act:AC9988CF-7835-4D01-85BB-1A89C221E7A1, *Aurelia columbia* sp. nov.: urn:lsid:zoobank.org:act:E8358C20-C234-4154-9434-548D81B6D888.

## Supplemental Information

Supplemental information for this article can be found online at http://dx.doi.org/10.7717/peerj.11954#supplemental-information.

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
