# Peer review of "The importance of molecular characters when morphological variability hinders diagnosability: systematics of the moon jellyfish genus Aurelia (Cnidaria: Scyphozoa)"

_PeerJ, doi:10.7717/peerj.11954_

## Round 0.1 · original submission · Minor Revisions

I now have reviews back from three well qualified referees who all agree that the manuscript is very well done and a worthy contribution to the field. Each also has a number of recommendations for minor revisions and how to improve the manuscript. Most are pretty straightforward, but one in particular seems of major consequence for revision: two of the referees ask for more detail on the morphological descriptions of these taxa, and how exactly the morphological characters line up with the newly erected species. As one referee states, they believe that a morphological description of any proposed species should be always provided, even if that morphological description is not necessarily diagnostic.

I know I have had this same discussion with several taxonomist colleagues, including some serving on the ICZN Commission, over the years, and I expect that others in addition to these referees will have the same questions with regard to your proposed taxonomy. How many taxonomic hypotheses based on a few ribosomal genes have been overturned when additional genomic information is brought to bear? Many of these folks I have spoken to have argued that there is little value in formally naming lineages, and whether the names used are legitimate or not under ICZN code is a fundamentally different issue than whether there is taxonomic benefit to erecting species names for every genetically detectable lineage. A similar tone comes through with the referees who ask for more morphological detail in the species descriptions. Many taxonomists I know will question if species can only be diagnosed through CAOS or QUIDDICH, what exactly what is the value of erecting a new species in such cases, and what do we gain by naming them? I feel that the comments raised by the referees are valid and representative of the field, so I look forward to seeing your response.

If you decide to undertake the suggested revisions, please ensure that all review comments (including those on the annotated manuscript) are addressed in a rebuttal letter that outlines exactly how you have addressed each comment. Any edits or clarifications mentioned in the rebuttal letter should also be inserted into the revised manuscript where appropriate. It is a common mistake to address reviewer questions in the rebuttal letter but not in the revised manuscript. If a reviewer raised a question, then your readers will probably have the same question so you should ensure that the manuscript can stand alone without the rebuttal letter. Directions on how to prepare a rebuttal letter can be found at: https://peerj.com/benefits/academic-rebuttal-letters/ if you need additional guidance.

I would also note here that it is PeerJ policy that additional references suggested during the peer-review process need only be included if the authors are in agreement that they are relevant and useful.

I look forward to seeing your revised manuscript and thank you for selecting PeerJ as the outlet for your work.

·

Basic reporting

This is such a big change for Aurelia, this paper will become a common reference, primarily Figure 10, that is why I feel this figure should be greatly improved:

- please significantly increase the resolution to avoid pixelation, I suggest this be one of the largest figures published.
- code existing, resurrected and new species (i.e. sp. 7 ... ) using a unique symbol to let the reader know these are new or not on the map.
- consider a different palette for colors, for example the in the Gulf of Mexico it is very difficult to tell the dark purple and dark brown points apart as they overlap. Also enabling jitter here should help.
- consider a slightly larger point size on the map (not the legend), points appear quite small relative to the map size.
- consider using the ggplot2 library if it was not used to create the plot

Experimental design

No comment.

Validity of the findings

No comment.

Additional comments

This paper is very well written, represents a major change for Aurelia and advances the field for scyphozoan taxonomy. I suggest minor revisions to the map figure before publication.

Reviewer 2 ·

Basic reporting

Lawley and coauthors provide a thorough revision of the iconic jellyfish genus Aurelia. They investigate the morphological and genetic diversity of several samples collected across the world ocean and provide new important data on this taxon. Specifically, they assess the morphological variability of the medusa stage in several species and perform statistical analyses to check for differences among species. At the same time, they produce genetic data based on four DNA regions to reconstruct a phylogentic hypothesis of the genus and to delineate species boundaries. Altogether they show that medusa morphology is not adequate to distinguish Aurelia species and find an alternative way to describe the species, that is the inclusion of genetic characters in the description.
The work is certainly well written and overall all analyses are well done. The stucture of the manuscript, together with figures, tables and supplemental files are adequate and almost ready for publication after minor revisions.
First of all, I suggest to shorten a bit the introduction by merging the first two paragraphs.

Experimental design

The scientific question and the performed morphological and molecular analyses are well explained. All analyses are rigorously done and consistent with the uptodate literature on the research field. I only have some suggestions that, in my opinion, could improve the scientific robustness of the work.
Regarding the morphological analyses I could suggest the authors to try to perform other analyses (such as a discriminant analysis) to assess the importance of each variable in characterising the species and to see if samples are correctly assigned to a given clade/species (a thing that should not expected in most cases, given the presented results).
Regarding molecular analyses, it should be advisable that phylogenetic analyses are performed using parsimony, maximum likelihood, and Bayesian inference for all datasets (single and multi-locus) in order to increase the robustness of the obtained phylogenetic hypotheses.
Moreover, authors could test their species hypotheses using some of the available DNA-based species delimitation techniques (including for instance ASAP, GMYC, PTP, etc).
Finally, I have nothing against naming cryptic species based on genetic data alone if morphology is not useful. Anyway, I believe that a morphological description of the proposed species should be always provided. Therefore, I suggest authors to provide a brief description of all the available life stages for all newly described species.

Validity of the findings

Overall, the work provides important insight into the diversity, evolution, and biogeography of Aurelia species and, in my opinion, fully deserves publication after the minor revisions suggest above.

Additional comments

no other comments

Reviewer 3 ·

Basic reporting

In general, I was able to follow and understand the text. The article follows the standard article structure in general. Raw data files including can be opened and these complemented the manuscript. The results are relevant to the hypotheses and the literature references are sufficient field context provided.

Experimental design

The research is original primary research within the Aims and Scope of the journal. The research question is well defined, relevant, and meaningful and it is stated how it fills an identified knowledge gap.

Validity of the findings

All underlying data have been provided; they are robust, statistically sound, and controlled. Additional details are requested (see general comments)

Additional comments

This research is relevant and fills an important knowledge gap and adds to our knowledge of biodiversity in the world.

In methodology and results should be add the appropriate tables for more details on the specimens.

To detail in methodology how are you present the species delimitation section in the results.

Add a comparative table with genetic and principal morphological features (e.g. morphology of rhopalia), besides the distribution of all species.

Add a section on possible morphological variability between female and male specimens.

Check more comments in the attached file.

Annotated reviews are not available for download in order to protect the identity of reviewers who chose to remain anonymous.

---

## Round 0.2 · accepted · Accept

Thank you for the detailed responses to the referee comments and for your careful revisions to the manuscript. All 3 referees were initially quite positive about the work, and you have addressed all their recommendations for improvement. Given your revisions, I am satisfied that all the feedback of the referees has been incorporated and that this version is improved by the process - I am confident the referees would be supportive as well. Therefore, I see no reason to further delay the process with additional review, and I am happy to move it forward into production.

Congratulations, and thank you for selecting PeerJ as the outlet for your work!